# PRINCIPLE PROCESS REWARD FOR SEARCH AGENTS

## ABSTRACT

Large Language Models (LLMs) increasingly rely on external tools such as search engines to solve complex agentic tasks that require reasoning and external knowledge retrieval. Recently, reinforcement learning with verifiable rewards (RLVR) has demonstrated its effectiveness in advancing capabilities of LLMs by rewarding the final answers via outcome rewards. While straightforward to supervise, outcome rewards only provide sparse signals and delayed feedback, which limits their effectiveness on long trajectories. Process rewards address this by evaluating intermediate steps, providing fine-grained supervision and encouraging grounded problem solving. However, it is notoriously hard to annotate step-wise labels, especially in non-verifiable process without "golden" answers. Furthermore, step-wise judgment requires the balance between local quality with contribution to the final outcome, as optimizing towards higher process reward may not always align with better final outcomes. To address the above challenges, we introduce Principle Process Reward (PPR), an RL approach that unifies principled step-level assessment and outcome verification. We train a principle-based reward model to improve the transparency and reliability of process evaluation, and further introduce a Reward Normalization (ReNorm) strategy to calibrate outcome and process rewards. Experiment results show that PPR achieves state-of-the-art performance across a wide range of benchmarks, demonstrating its impressive robustness and generalization. Our code and model collection is available in this link.

## 1 INTRODUCTION

Large Language Models (LLMs) have achieved remarkable progress across a wide range of tasks, from open-domain question answering to multi-step reasoning (Guo et al., 2025; OpenAI, 2025b; Comanici et al., 2025). A key factor for success is their abilities to leverage external tools such as search engines, calculators, code interpreters, and browsers (DeepMind, 2025; Guo et al., 2024; OpenAI, 2025a). In particular, the search engine is a linchpin tool that provides verifiable and up-to-date knowledge for LLMs, helping to ground their answers and reduce hallucinations. However, training LLM agents to leverage tools effectively still remains challenging, as the complex behavior involving task decomposition, query generation, information aggregation, and stopping decisions.

Reinforcement Learning with Verifiable Rewards (RLVR) has demonstrated notable success in enhancing the reasoning capabilities of LLMs (Jaech et al., 2024; Qwen, 2025), However, when it comes to agentic scenarios, the reward landscape is too sparse for effective model training via conventional RLVR. Most existing RL-based methods for tool-use still rely on evaluating verifiable outcomes against golden answers and compute outcome rewards as the exclusive supervision (Qian et al., 2025; Yu et al., 2024). While straightforward, outcome rewards lack mechanisms for precise credit assignments across intermediate steps (Cui et al., 2025). This problem is especially severe in agentic tasks with long trajectories before observing final answers (Xiong et al., 2024).

Process Reward Models (PRMs), which provide richer reward signals by evaluating intermediate steps (Lightman et al., 2023; Zhang et al., 2025), encourage more grounded problem-solving strategies and make credit assignments easier. It has demonstrated effectiveness in domains like math and coding (Liu et al., 2025b; Zou et al., 2025; Khalifa et al., 2025b). However, adopting PRMs in agentic search remains challenging: First, the intermediate steps, such as invoking a search engine, are inherently non-verifiable due to the lack of gold answers. Second, optimizing for process rewards does not necessarily translate into better final outcomes (Guo et al., 2025). As a result, an elaborate strategy to integrate and balance local process fidelity and overall task success is necessary.

To address these challenges, we propose **Principle Process Reward (PPR)**, a RL framework for agentic tasks that involves a hybrid reward with principle-based process evaluation and outcome verification. PPR is built upon two key innovations: First, we develop a **Principle-based Process Reward Model (PPRM)** that grounds step-wise judgments in interpretable, generalizable principles. Instead of relying on ad-hoc heuristics, PPRM leverages a set of pre-defined principles-such as correctness, relevance, and consistency-and learns to adapt them into context-sensitive rubrics tailored to each trajectory. This principle-driven design improves the transparency, stability and reliability of process rewards. It also enhances robustness to non-verifiable and ambiguous intermediate actions. Second, we introduce **ReNorm**, a reward normalization method which unifies and calibrates outcome and process rewards before advantage estimation. A naive combination of outcome and process feedback can destabilize optimization especially in long-horizon tasks where local fidelity and global success may conflict. ReNorm addresses this by rescaling reward magnitudes, balancing short-term process fidelity against long-term task success. This prevents reward hacking and enables stable and scalable RL training for agents operating over long trajectories. Together, PPR provides a general training recipe for aligning LLMs with non-verifiable agentic tasks. By combining step-level guidance with calibrated outcome signals, PPR produces interpretable reasoning which enables safer and endorsed alignment with LLMs.

We evaluate PPR extensively on both in-domain and out-of-domain benchmarks in search agent tasks including General QA and Multi-Hop QA. PPR consistently outperforms all baselines by up to 28% average relative improvement over non-RL baselines and 15% over previous RL methods counterparts. Beyond this, we also conduct a wide range of analysis for our critical designs including principles and ReNorm. PPR demonstrates its impressive robustness in training stability and highlights the potential for scalability and generalization in tool-use agents. Additionally, to demonstrate the effectiveness of our PPRM, we built a Non-Verifiable Benchmark (NVProcessBench) for evaluating PRMs' effectiveness in agentic tasks. Our main contributions can be summarized as:

1. We introduce PPR, an RL framework that unifies outcome verification with a principled, context-adaptive process reward for search agents.

2. We design and train a PPRM that evaluates intermediate actions against general principles and adapt to sample-specific rubrics.

3. We propose an elaborate ReNorm strategy that integrates and normalize process and outcome rewards, which greatly stabilizes RL training and effectively prevents collapse in long-trajectory optimization.

4. We construct NVProcessBench, a benchmark for evaluating reward models on non-verifiable processes with the data collected from real-world trajectories.

## 2 Related Works

### 2.1 Large Language Models and Search Agents

To ensure access to the most up-to-date information, recent LLMs have been enhanced to interact more efficiently with search engines (Anthropic, 2025; Yang et al., 2025). Training-free pipelines such as retrieval-augmented generation (RAG) (Shao et al., 2023) and its variants (Trivedi et al., 2022a; Shao et al., 2023; Li et al., 2025b) typically involve a round of search queries and append the retrieved information to the subsequent context. Although effective, RAG-based methods are difficult in fine-grained and efficient interaction. This occurs potential irrelevant and long-tailed retrieval, which in turn increases computational overhead for LLMs.

RL has emerged as a promising paradigm to improve the capabilities of LLMs (Jaech et al., 2024; Guo et al., 2025), and recent studies have explored its application to search agents. For example, Jin et al. (2025) adopts a rule-based RL recipe to train models to generate search queries during multi-turn rollouts. Chen et al. (2025) combines outcome rewards with formatting constraints to encourage effective reasoning with retrieved knowledge. Sun et al. (2025b) leverages another LLM to simulate the search engine, thereby reducing search costs. However, these methods merely rely on outcome answers by exact match or F1 score, without supervision on intermediate reasoning steps. Such sparse feedback complicates credit assignment, destabilizes RL training, or even reward hacking (Cui et al., 2025; Amodei et al., 2016). Instead, we propose a paradigm that explicitly supervise intermediate steps and coordinates them with final outcomes by a normalization.

## 2.2 Reinforcement Learning with Process Reward

Process reward enrich the reward function by providing denser rewards comparable with outcome rewards. This reward shaping technique has shown benefits including guiding best-of-N sampling at test time (Lightman et al., 2023), facilitating high-quality data selection (Zou et al., 2025), and enabling process-level supervision in LLM training (Setlur et al., 2024; Zhang et al., 2025). However, designing effective process reward mechanisms is inherently challenging. A main concern is that obtaining reliable process labels is challenging especially for non-verifiable reasoning steps. It requires substantial human annotation to create high-quality labels for the process, making it costly and time-consuming. Previous works have explored Monte Carlo (MC) rollouts to estimate step-level Q-values (Wang et al., 2023) and implicit token-level reward estimation by jointly training a reward model with the policy model (Cui et al., 2025; Zhang et al., 2024b) to mitigate this issue. While effective for mathematical reasoning, these approaches show limited generalization across agentic domains (Zheng et al., 2024; Luo et al., 2024). Another challenge is hybrid reward modeling can cause reward hacking and instability (Guo et al., 2025). Prior work has primarily focused on regulating process reward using outcome signals (Zhang et al., 2024a; Gao et al., 2024), but lack of exploration in harmonizing at a step-grained level. In this work, we develop a principle process reward model that leverages explicit reasoning principles (Kimi, 2025; Liu et al., 2025b) to provide structured feedback and verification without large-scale training. Furthermore, we introduce a novel reward normalization method to align process and outcome rewards, which ensures training stability. We demonstrate the effectiveness of our approach on agentic search tasks and highlight its possible scalability towards broader agentic reasoning problems.

## 3 Method

In this section, we introduce Principle Process Reward (PPR), our proposed hybrid RL paradigm for non-verifiable search agents. We begin by formulating the reinforcement learning objective with search agents (Sec. 3.1), then present our reward design (Sec. 3.2), and finally discuss the training detail of our Principle Process Reward Model (PPRM) used in PPR (Sec. 3.3).

### 3.1 Overview and Optimization Methods

As illustrated in Figure 1, given a query $q$ drawn from the dataset $\mathcal{D}$, the policy LLM $\pi_\theta$ can interact with search engine $\mathcal{S}$ over $n$ turns. At each turn $t < n$, $\pi_\theta$ generates a reasoning segment $R_t$ and search query $S_t$ if needed. Search engine $\mathcal{S}$ returns retrieved information $\text{info}_t$, and we update current context to $c_{t+1} = (c_t, R_t, S_t, \text{info}_t)$ for the next action, until it generates the final answer $O$ or reach to max turn limit. Thus the complete trajectory can be presented as $\tau = (R_1, S_1, \text{info}_1, \ldots, R_{n-1}, S_{n-1}, \text{info}_{n-1}, O)$. The RL objective is as follows:

$$\max_{\pi_\theta} \mathbb{E}_{q \sim \mathcal{D},\, \tau \sim \pi_\theta(\cdot|q;\mathcal{S})}[r_\phi(q, \tau)] - \beta \, \mathbb{D}_{\text{KL}}\big[\pi_\theta(\tau \mid q; \mathcal{S}) \,\|\, \pi_{\text{ref}}(\tau \mid q; \mathcal{S})\big], \quad (1)$$

where $\pi_{ref}$ is the reference LLM, and $\mathbb{D}_{\text{KL}}$ is KL-divergence measure. In reward function $r_\phi$, the final answer $O$ derived from the last turn is calculated for the outcome reward scaler $r_o$, and a PRM evaluates step-wise reward under certain principles and outputs a reward scalar, which is placed at the end of the corresponding place in the process-reward tensor $\mathbf{r}_p$ after the reward normalization. Final reward score tensor $\mathbf{r}$ will be aggregated with reference model for advantage estimation.

We compute advantages with Generalized Advantage Estimation (GAE) (Schulman et al., 2015) and optimize with Proximal Policy Optimization (PPO) (Schulman et al., 2017). In practice, we optimize the clipped PPO surrogate at the token level over model-generated tokens, and mask the retrieved tokens indicated in $\text{info}$. The optimization maximizes

$$\mathcal{L}_{\text{clip}}(\theta) = \mathbb{E}_{q \sim \mathcal{D},\, y \sim \pi_\theta(\cdot|c_t)}\left[\frac{1}{\sum_{t=1}^{|y|} I(y_t)} \sum_t I(y_t) \min\Big(\frac{\pi_\theta(y_t|c_t)}{\pi_{\text{old}}(y_t|c_t)} A_t,\, \text{clip}\big(\frac{\pi_\theta(y_t|c_t)}{\pi_{\text{old}}(y_t|c_t)}, 1 - \epsilon, 1 + \epsilon\big) A_t\Big)\right], \quad (2)$$

where $I(y_t) \in \{0, 1\}$ mask out retrieved tokens, and the advantage estimation $A_t$ is calculated by GAE, together with a value regression: $\mathcal{L}_{\text{value}}(\phi) = \mathbb{E}[(V_\phi(c_t) - (A_t + V_\phi(c_t)))^2]$.

### 3.2 Reward Design

In this section, we discuss the details of reward design in PPR. The core insight of PPR is a hybrid reward with process and outcome supervision through a normalization. We first describe the out-

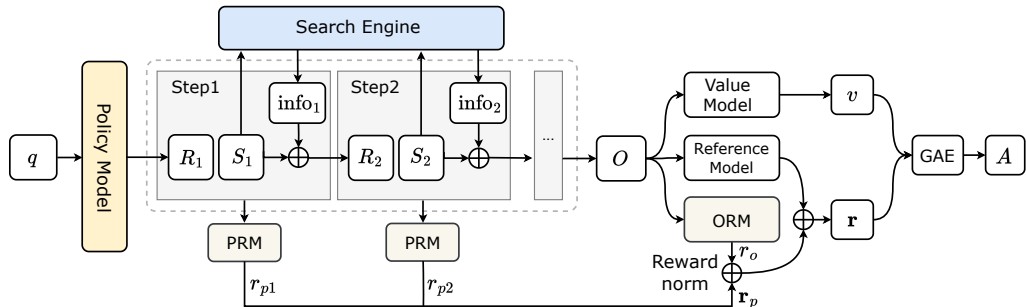

Figure 1: Overview of PPR Rollout: Given a user query $q$, the policy model interacts with a search engine and produces multi-step conversations. At $t$-th step, it generates a reasoning trace $R_t$ and a search query $S_t$ if applicable. The retrieved information $\text{info}_t$ is appended to the context for subsequent steps. PRM will assign step-wise credits while ORM will assess the final answer $O$.

come reward used in PPR, then present the process reward and its principle design, and finally our reward normalization strategy.

**Outcome Reward.** Similar to Jin et al. (2025), we calculate the outcome reward by exact match:

$$r_o = \text{ORM}(q, \pi, O) = \text{EM}(O, O_{\text{golden}}) = \mathbb{I}\{O = O_{\text{golden}}\} \in \{0, 1\}, \tag{3}$$

which provide supervision towards LLM's final answer $O$ strictly.

**Principle Design and Principle Process Reward.** Unlike math, where intermediate steps may be directly verifiable, search trajectories are difficult to judge step by step due to the lack of process annotations. We cannot rely on golden labels for process reward calculation, since no definitive reference exists for such non-verifiable processes. Inspired by Khalifa et al. (2025a), we recast step-wise judgment of non-verifiable trajectories as a quantitative instruction-following task. To evaluate it, we further develop a benchmark that quantifies the performance of reward models on such trajectories (Sec. 3.3). We construct a principle set $\mathcal{P} = \{p_1, \ldots, p_K\}$, covering criteria such as formatting, correctness, and alignment that characterize a well-formed process. Reward models are required to adhere to these principles and generate concise rubrics for fine-grained evaluation, which summarize to `<analysis>`. Building on this, we fine-tune a principle-based generative reward model (PPRM) on Qwen3-8B. At each step, PPRM dynamically refers to the relevant principles from $\mathcal{P}$ to ground its analysis and generate an interpretable score on a standardized scale.

As shown in Figure 2, we create an explicit QA pair $\left(Q_t^{\text{prm}}, Y_t^{\text{prm}}\right)$ for the $t$-th step as follows:

$$Q_t^{\text{prm}} = (q, R_1, S_1, \text{info}_1, \ldots, R_{t-1}, S_{t-1}, \text{info}_{t-1}), \quad Y_t^{\text{prm}} = (R_t, S_t, \text{info}_t), \tag{4}$$

where the initial $Q_1^{\text{prm}}$ is the origin user query $q$. PPRM calculates the process-supervision reward score by evaluating $\left(Q_t^{\text{prm}}, Y_t^{\text{prm}}\right)$ under certain principles from the entire set:

$$\forall t \in \{1, \ldots, n-1\}: \quad \hat{r}_{p,t} = \frac{\sum_{p_i \in \mathcal{P}_t} \texttt{<score>}_t(p_i)}{\sum_{p_i \in \mathcal{P}_t} \texttt{<max\_score>}(p_i)} \in [0, 1], \mathcal{P}_t \subseteq \mathcal{P} \tag{5}$$

where PPRM are instructed to generate a final score indicated by `<score>` and `<max_score>` based on `<analysis>`. The process reward scaler will be added on the corresponding position at the end of each turn in the reward tensor. The details of PPRM data collection and training are discussed in Section 3.3 and representative principles for PPRM are shown in Appendix A.7.

**ReNorm.** Consistent with prior findings (Cheng et al., 2025; Rafailov et al., 2024), we also observe that combining outcome and process rewards leads to severe instability during RL training. To mitigate this issue, we propose Reward Normalization (ReNorm) to unify and calibrate the discrete, sparse outcome reward and continuous, dense process reward. Specifically, we first broadcast the outcome reward to every step and centralize the aggregated reward signals to ensure step-wise credit is anchored to end-task correctness while keeping magnitudes comparable across different turns. ReNorm performs:

$$r = (r_p - \mu) + (r_o - \pi), \tag{6}$$

Figure 2: Reward tensor. The PPRM generates step-wise rewards $\hat{r}_{p_i}$ by dynamically selecting relevant principles from the principle set according to the given context. For example, in the first turn PPRM selects #1, #2, and #3, while in the second turn it selects #1, #3, and #4. A rule-based outcome reward $r_o$ is computed from the final response. ReNorm $f$ normalizes $\hat{r}_{p_i}$ and $r_o$ to obtain the final step reward $r_{p_i}$. All rewards are inserted into the reward tensor at their corresponding positions, with all remaining entries set to zero.

---

**Algorithm 1** Rollout with Principle-based Process Reward and Reward Normalization

---

**Require:** input query $q$, trajectory $\tau$, policy model $\pi_\theta$, max turns $n$, principle library $\mathcal{P}$, search engine $\mathcal{S}(\cdot)$, per-step reward score $\{r_t\}_{t=1}^n$, final answer $O$.

1: $\tau \leftarrow \emptyset; t \leftarrow 0$          ▷ initialize trajectory and step count
2: **while** $t < n$ **do**
3:      **Policy action:** $(R_t, S_t) \sim \pi_\theta(\cdot \mid q; \mathcal{S})$          ▷ get search query
4:      $\text{info}_t \leftarrow \mathcal{S}(S_t)$          ▷ retrieve information
5:      $\tau \leftarrow \tau \cup \{R_t, S_t, \text{info}_t\}$
6:      **PPRM:** for $p_i \in \mathcal{P}_t \subseteq [\mathcal{P}], \hat{r}_{pt} \leftarrow \frac{\sum_{p_i \in \mathcal{P}_t} \texttt{<score>}_i}{\sum_{p_i \in \mathcal{P}_t} \texttt{<max\_score>}_i}$
7:      **if** `<answer>` detected in $\tau$ **then**
8:          return final answer $O$; $r_o \leftarrow \mathbb{I}\{O = O_{\text{golden}}\}$
9:      **end if**
10:     $t \leftarrow t + 1$
11: **end while**
12: **ReNorm**: $r_{p,t} \leftarrow \hat{r}_{p,t} + r_o - 1 \quad \forall t \in \{1, \ldots, n-1\}$

---

where $\pi = \mathbb{E}[r_o]$ and $\mu = \mathbb{E}[r_p]$, thus ReNorm unifies $\mathbb{E}[r] = 0$. In practice, we assume data distribution unified thus $\mu \approx \pi \approx \frac{1}{2}$:

$$\forall t \in \{1, \ldots, n-1\} : r_{p,t} = \text{ReNorm}(\hat{r}_{p,t}, r_o) = \hat{r}_{p,t} + r_o - 1. \in [-1, 1] \tag{7}$$

Intuitively, when the final answer is wrong ($r_o = 0$), the process reward ($r_{pt}$) is non-positive; when correct ($r_o = 1$), it is non-negative. ReNorm enforces sign consistency which assigns no positive credit on failed trajectories. The bounded range stabilizes Temporal-Difference (TD) errors and GAE/PPO updates, and centering reduces baseline-leakage bias and advantage variance—thereby discouraging process-only "score inflation" that does not improve correctness. We provide more details in Appendix A.8 and provide an ablation with different normalization in Section 4.3.

As indicated in Figure 2, with the process reward after ReNorm, reward score $r_i$ at each token $i$ can be presented as follows:

$$r_i = \begin{cases} r_{pt}, & \text{if } i \text{ is the end token of turn } t \text{ when } t < n, \\ r_o, & \text{if } i \text{ is final token of the entire trajectory}, \\ 0, & \text{Otherwise.} \end{cases} \tag{8}$$

## 3.3 PPRM TRAINING

To train PPRM, we first sampled approximately 2k multi-turn search inference trajectories. To ensure high data quality, we applied a two-stage filtering pipeline: GPT-5 was used as an automatic verifier to discard unanswerable, incorrect, or low-quality responses, followed by human review to

further eliminate low quality samples. The resulting dataset was used to perform supervised fine-tuning (SFT) of PPRM, initialized from Qwen3-8B.

Existing PRM benchmarks have primarily focused on verifiable domains such as mathematical reasoning (Zhang et al., 2025). To evaluate performance in more challenging settings, we constructed NVProcessBench, a new benchmark targeting non-verifiable, multi-turn trajectories, particularly in search-based tasks. NVProcessBench consists of roughly 2k carefully curated examples, each annotated with binary step-level correctness labels. Full details are provided in Appendix A.4.

Evaluation results are shown in Table 1. As baselines, we consider both Bradley–Terry–style models (Skywork-Reward-V2-Llama-3.1-8B Liu et al. (2025a)) and GRMs (Qwen3-8B Yang et al. (2025), Qwen3-235B-A22B, ThinkPRM-7B Khalifa et al. (2025b)). PPRM achieves 0.613 accuracy, demonstrating strong capability in identifying correct steps and robustness against fluctuations in negative predictions. Notably, Qwen3-235B-A22B frequently misidentifies its role as a reward model, instead acts like an actor model and directly generates answers rather than providing evaluations. While larger models often have better overall performance, they are not consistently superior for structured outputs (Zhou et al., 2024). An ablation study of this effect is presented in Section 4.3.

Table 1: PPRM vs. baselines on NVProcessBench. Qwen3-8B and Qwen3-235B-A22B serve as general-purpose generative reward models. Skywork-Reward-V2-Llama-3.1-8B (Skywork-V2) is a Bradley–Terry style reward model designed to score question–answer pairs. ThinkPRM is a generative reward model fine-tuned specifically for reasoning tasks.

| Process Reward Models | Accuracy |
|---|---|
| Qwen3-8B | 0.590 |
| Qwen3-235B-A22B | 0.116 |
| Skywork-V2 | 0.559 |
| ThinkPRM | 0.242 |
| PPRM (Ours) | **0.613** |

## 4 EXPERIMENTS

### 4.1 IMPLEMENTATION DETAILS

**Experiment setup.** We use the 2018 Wikipedia dump Karpukhin et al. (2020) as the knowledge base and E5 Wang et al. (2022) as the retriever. Our base policy model are initialized from Qwen2.5-3B (Base/Instruct) and Qwen2.5-7B (Base/Instruct) (Yang et al., 2025). The Instruct models are finetuned from the Base models with augmentation on instruction following abilities. We adopt veRL (Sheng et al., 2025) with FSDP (Zhao et al., 2023) as the training engine and vLLM (Kwon et al., 2023) as the inference engine. We use exact match (EM) as the metric, which is used both in our RL recipe and evaluation. Details of parameters can be found in Appendix A.3.

**Datasets and Benchmarks.** Our training datasets span both the general question answer dataset NQ (Kwiatkowski et al., 2019) and the multi-hop question answer dataset HotpotQA (Yang et al., 2018). To demonstrate our method's generalization, we evaluate it both on in-domain data distribution (ID) and out-of-domain data distribution (OOD). (1) ID data: NQ, HotpotQA (2) OOD data: TriviaQA (Joshi et al., 2017), PopQA (Mallen et al., 2022), 2WikiMultiHopQA (Ho et al., 2020), Musique (Trivedi et al., 2022c), Bamboogle (Press et al., 2022). These datasets encompass a diverse range of search domains that are fundamental for agentic tasks. To ensure fairness, all the baselines are trained and evaluated within the same dataset categories.

**Baselines.** We compare a wide range of baselines including the following: (1) *Inference without RL*: Direct Inference (DI), Chain-of-Thought (CoT) reasoning (Wei et al., 2022), Retrieval-Augmented Generation (RAG) (Lewis et al., 2020), IRCoT (Trivedi et al., 2022b), Iter-RetGen (Shao et al., 2023), and Search-o1 (Li et al., 2025a). (2) *RL with ORMs*: Search-R1 (Jin et al., 2025), R1-Searcher (Song et al., 2025), ZeroSearch (Sun et al., 2025a), and ReSearch (Chen et al., 2025). (3) *RL with PRMs*: To demonstrate the effectiveness of our method, we adopt other generative models as the PRMs in our recipe, including Bradley-Terry based Skywork-Reward-V2-Llama-3.1-8B (Liu et al., 2025a), Qwen3-8B and Qwen3-235B-A22B (Yang et al., 2025). To ensure fair comparison, we reproduced all the baselines under the same experiment settings as PPR. The GRMs also share the same prompts and principles exhibited in Appendix A.7.

---

[1] We use Skywork-Reward-V2 as the process reward model, denoted simply as 'Skywork-V2' in the table. The same applies to Qwen3-8B and Qwen3-235B-A22B.

Table 2: Main results on General QA and Multi-Hop QA benchmarks. The baselines are grouped into three categories: non-RL methods, RL with outcome-only rewards, and RL with process rewards. **Bold** indicates the best performance, and underline denotes the second best.

| Method | General QA | | | Multi-Hop QA | | | | |
|---|---|---|---|---|---|---|---|---|
| | **NQ** | **TriviaQA** | **PopQA** | **HotpotQA** | **2wiki** | **Musique** | **Bamboogle** | **Avg.** |
| **Qwen2.5-3b-Base/Instruct** | | | Without Reinforcement Learning | | | | | |
| DI (inst) | 0.122 | 0.322 | 0.130 | 0.160 | 0.248 | 0.022 | 0.024 | 0.147 |
| CoT | 0.002 | 0.017 | 0.001 | 0.008 | 0.248 | 0.000 | 0.000 | 0.039 |
| RAG | 0.340 | 0.560 | 0.404 | 0.263 | 0.231 | 0.052 | 0.072 | 0.275 |
| IRCoT | 0.238 | 0.443 | 0.329 | 0.262 | 0.190 | 0.065 | 0.200 | 0.247 |
| Iter-RetGen | 0.358 | 0.575 | 0.419 | 0.280 | 0.240 | 0.062 | 0.120 | 0.293 |
| Search-o1 | 0.201 | 0.310 | 0.173 | 0.231 | 0.208 | 0.045 | 0.208 | 0.197 |
| | | | Reinforcement Learning with Outcome Reward | | | | | |
| Search-R1 | 0.353 | 0.552 | 0.382 | 0.323 | 0.310 | 0.104 | 0.266 | 0.327 |
| ZeroSearch | 0.392 | **0.605** | 0.417 | 0.287 | 0.284 | 0.072 | 0.144 | 0.314 |
| ReSearch | 0.366 | 0.541 | 0.375 | 0.276 | 0.264 | 0.068 | 0.104 | 0.285 |
| | | | Reinforcement Learning with Process Reward | | | | | |
| Skywork-V2 [1] | 0.314 | 0.439 | 0.293 | 0.320 | 0.330 | 0.110 | 0.242 | 0.292 |
| Qwen3-8B | 0.351 | 0.533 | 0.383 | 0.341 | **0.346** | 0.108 | 0.242 | 0.329 |
| Qwen3-235B-A22B | 0.335 | 0.527 | 0.373 | 0.350 | 0.319 | 0.133 | **0.339** | 0.339 |
| **PPR-Instruct (ours)** | **0.441** | 0.594 | **0.424** | **0.390** | 0.342 | **0.150** | 0.282 | **0.375** |
| **PPR-Base (ours)** | 0.423 | 0.565 | 0.411 | 0.353 | 0.340 | 0.127 | 0.280 | 0.357 |
| **Qwen2.5-7b-Base/Instruct** | | | Without Reinforcement Learning | | | | | |
| DI (inst) | 0.142 | 0.437 | 0.152 | 0.193 | 0.255 | 0.037 | 0.104 | 0.189 |
| CoT | 0.124 | 0.416 | 0.150 | 0.178 | 0.255 | 0.037 | 0.088 | 0.178 |
| RAG | 0.341 | 0.590 | 0.395 | 0.302 | 0.237 | 0.056 | 0.216 | 0.305 |
| IRCoT | 0.237 | 0.516 | 0.333 | 0.258 | 0.150 | 0.053 | 0.216 | 0.252 |
| Iter-RetGen | 0.366 | 0.610 | 0.415 | 0.326 | 0.257 | 0.073 | 0.208 | 0.322 |
| Search-o1 | 0.152 | 0.345 | 0.210 | 0.177 | 0.178 | 0.077 | 0.312 | 0.207 |
| | | | Reinforcement Learning with Outcome Reward | | | | | |
| Search-R1 | 0.418 | 0.577 | 0.396 | **0.394** | 0.307 | 0.161 | 0.379 | 0.376 |
| R1-Searcher | 0.338 | 0.521 | 0.343 | 0.365 | **0.399** | **0.173** | 0.384 | 0.360 |
| ReSearch | 0.366 | 0.605 | 0.391 | 0.378 | 0.386 | 0.166 | 0.376 | 0.381 |
| | | | Reinforcement Learning with Process Reward | | | | | |
| Skywork-V2 | 0.391 | 0.576 | 0.394 | 0.323 | 0.238 | 0.113 | 0.348 | 0.326 |
| Qwen3-8B | 0.379 | 0.572 | 0.388 | 0.337 | 0.274 | 0.130 | 0.363 | 0.349 |
| Qwen3-235B-A22B | 0.340 | 0.553 | 0.345 | 0.324 | 0.272 | 0.126 | **0.419** | 0.340 |
| **PPR-Instruct (ours)** | **0.467** | **0.625** | **0.450** | 0.387 | 0.310 | 0.155 | 0.412 | **0.400** |
| **PPR-Base (ours)** | 0.458 | 0.610 | 0.437 | 0.386 | 0.355 | 0.147 | 0.355 | 0.390 |

## 4.2 MAIN RESULTS

The main results comparing PPR with baseline methods across General QA and Multi-Hop QA datasets are shown in Table 2. Overall, PPR consistently outperforms all baselines. With 3B and 7B models, it achieves 28% and 24% average relative improvement over non-RL baselines, 15% and 5% over RL methods with outcome-only rewards, and 11% and 15% improvements over RL training with existing LLM-as-PRM counterparts. These gains hold across both in-distribution and out-of-distribution evaluation. Furthermore, we observe that PRMs without principled design and proper normalization method underperform outcome-only approaches, highlighting the necessity of both principle-based reward model and the ReNorm strategy for effective training (see Sec. 4.3 for detailed analysis). Additionally, PPR is effective across different model variants, achieving strong performance on both 3B/7B and Base/Instruct settings, with larger models consistently demonstrating greater learning ability.

## 4.3 ANALYSIS

**Learning Stability.** Training stability is a critical challenge in RL, as unstable optimization may lead to reward collapse and training failure. Our proposed PPR method improves both stability and training performance by providing dense, principle-based feedback that prevents divergence during long-trajectory optimization. In Figure 3a we visualize the training rewards over increasing steps on NQ dataset, comparing PPR with existing baselines (*i.e.,* Search-R1, using Skywork-V2 and Qwen3-8B to generate process reward) on Qwen2.5-3B-Instruct. It shows that training rewards with PPR

increase steadily and consistently, whereas the compared baselines exhibit declining performance or even collapse as training progresses.

**Reward Design Methods.** A key challenge in process-reward RL is balancing process and outcome rewards. Without a carefully designed integration strategy, PR can introduce instability and noise, severely degrading the performance. To investigate this, we compare different PR credit assign and regulation strategies: (1) w/o Norm—directly assign raw process rewards at each step without normalization, (2) Rescale Norm—scaling the sum of process rewards in a trajectory to [0,1], (3) Reduced Reward-directly summation of all process rewards in a trajectory on outcome reward, (4) our proposed ReNorm. As shown in Figure 3b, ReNorm yields stable and steadily improving training rewards, while other normalization strategies may cause the training to collapse. Table 3 further confirms these findings, showing that ReNorm combined with a dedicated PRM delivers the strongest test-set performance, underscoring their importance in stable and effective RL training.

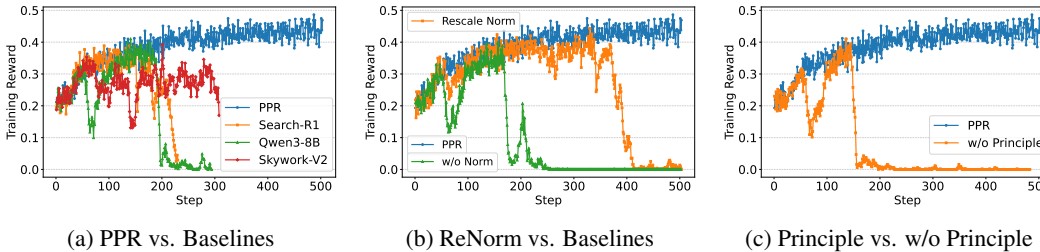

|                       |                          |                               |
| :-------------------: | :----------------------: | :---------------------------: |
| (a) PPR vs. Baselines | (b) ReNorm vs. Baselines | (c) Principle vs. w/o Principle |

Figure 3: Training rewards Qwen2.5-3B-Instruct on the NQ dataset. (a) PPR vs. Baselines: PPR consistently achieves the highest rewards, while ORM-based (Search-R1) and PRM-based (Qwen3-8B, Skywork-V2) baselines collapse before 300 steps or exhibit severe fluctuations; (b) ReNorm vs. baselines: ReNorm outperforms other normalization methods in both performance and stability; (c) Principle vs. w/o Principle: PPR demonstrates the principle's effectiveness in PRM design.

Table 3: Ablation on principles and reward design in PPR. 'w/o Principle' removes the principle design. 'Reduced Reward' adds process rewards to the final outcome reward. 'w/o Norm' computes advantages directly by outcome and process rewards without normalization. 'Rescale Norm' scales the sum of all process rewards in one trajectory equals to the scale of outcome rewards. With both principle design and ReNorm, PPR consistently outperforms all other baselines.

| Method         | NQ        | TriviaQA  | PopQA     | Avg.      |
| :------------- | :-------- | :-------- | :-------- | :-------- |
| w/o Principle  | 0.278     | 0.468     | 0.320     | 0.355     |
| Reduced Reward | 0.367     | 0.568     | 0.404     | 0.446     |
| w/o Norm       | 0.359     | 0.553     | 0.240     | 0.384     |
| Rescale Norm   | 0.377     | 0.548     | 0.384     | 0.436     |
| ReNorm(Ours)   | **0.441** | **0.594** | **0.437** | **0.491** |

**Valid Judge Rate.** As Table 2 shows, our trained RM tailored to the task consistently outperforms general-purpose LLMs, like Qwen3-8B and Qwen3-235B-A22B. One reason is that it distills the capabilities of larger models into a specialized verifier, achieving performance comparable to models with hundreds of billions of parameters. Another reason is its ability to produce stable, valid judge scores throughout training, which is crucial since failures in formatting or principle-level scoring can lead to persistently low rewards and eventual collapse of RL training. As shown in Figure 4a, our fine-tuned RM maintains a near-perfect valid judge rate, whereas Qwen3-8B and Qwen3-235B-A22B frequently fail. Notably, the 235B model performs even worse than the smaller 8B model, suggesting that larger models are not necessarily better verifiers; their broader exploration space often hinders strict instruction following. This explains why even powerful LLMs may fail to guide training effectively, as also reflected in Table 1.

**Response Length and Valid Search.** We analyze the Qwen2.5-3B-Instruct model to study the dynamics of response length and the number of valid search engine calls during training. As shown in Figure 4c and Figure 4b, the model initially increases both response length and search frequency, which corresponds to a sharp rise in training reward. Over time, it converges to issuing a single valid search call per query—consistent with the characteristics of the dataset.

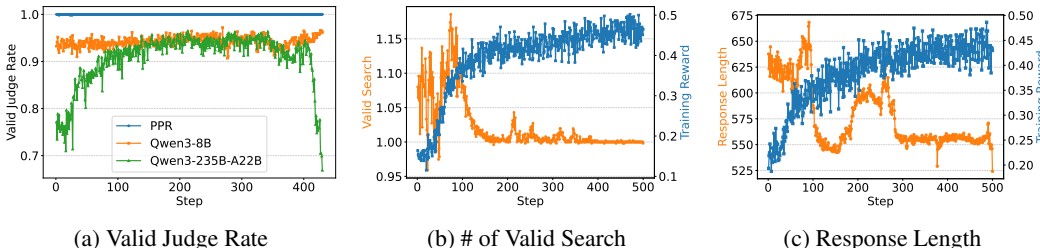

(a) Valid Judge Rate      (b) # of Valid Search      (c) Response Length

Figure 4: Analysis on (a) Valid Judge Rate: PPR consistently achieves perfectly following designed format, while baselines achieve 0.7 to 0.9 success rates. (b) # of Valid Search: PPR avoids generating redundant or invalid queries and stabilize over training. (c) Response Length: The average response length converges over training.

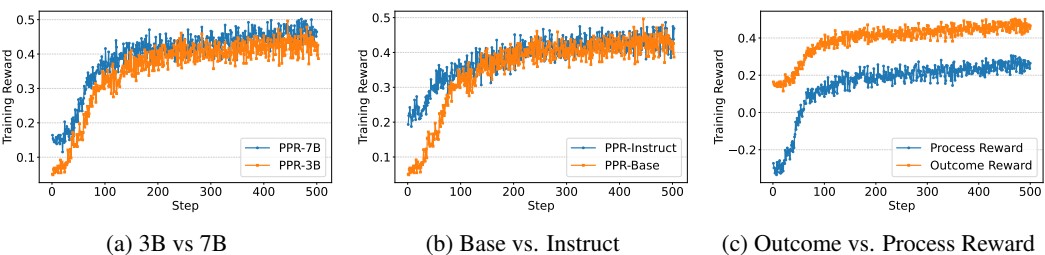

(a) 3B vs 7B      (b) Base vs. Instruct      (c) Outcome vs. Process Reward

Figure 5: (a) Qwen2.5-3B vs. Qwen2.5-7B: PPR has demonstrated scalability regarding the model sizes. (b) Qwen2.5-3B vs. Qwen2.5-3B-Instruct: Base models initially starts worse but converges similar to Instruct models. (c) Outcome vs. Process Reward: Process and outcome reward can align with each other in PPR, indicating process supervision follows final judgment and benefits learning.

**Variants of Base Models.** We also compare the training performance of different variants of base models. In Figure 5a, both Qwen2.5-3B and 7B achieve stable reward improvement on the NQ dataset, with the 7B model consistently outperforming the 3B model due to its larger capacity. In Figure 5b, Qwen2.5-3B-Base initially lags behind its Instruct counterpart but catches up after RL training, achieving comparable performance. This suggests that while instruction tuning accelerates early learning in reasoning by enhanced instruction following ability, RL can effectively close the gap that enabling base models to achieve comparable performance. Overall, the results indicate that our method is robust across both model scales and initialization types, consistently enabling effective learning.

**Outcome and Process Reward.** We analyze the Qwen2.5-3B-Instruct model to compare the evolution of process and outcome rewards during training. As shown in Figure 5c, the process reward increases rapidly at the beginning, providing dense feedback that guides the model's early exploration. The outcome reward then improves as a consequence of this guidance, and eventually both rewards rise in a synchronized and stable manner. This dynamic suggests that the PRM plays a critical role in shaping intermediate behaviors especially in the begin of training, which in turn stabilizes training and prevents collapse.

## 5 CONCLUSION

In this work, we propose PPR, an RL approach that incorporates step-level assessment into the training of LLM agents for multi-turn tool use. To address the challenge of non-verifiable step evaluation, we design a principle-based process reward, and introduce ReNorm to integrate process and outcome rewards. We validate PPR on search tasks, showing that it consistently outperforms strong baselines across diverse datasets in both in-domain and out-of-domain settings. Our analysis demonstrates that the principle-based PRM generates high-quality process rewards, while ReNorm stabilizes training and mitigates reward hacking. In addition, we introduce NVProcessBench, the first benchmark for non-verifiable process evaluation, and will release a full version in the future to facilitate further progress by the community. In conclusion, we validate the effectiveness of PPR in search tasks and highlight its potential as a general framework for stable and effective RL training, laying the foundation for future works to more complex agentic tool-use tasks.

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

# A APPENDIX

## A.1 LARGE LANGUAGE MODEL USAGE

We leveraged LLMs to polish the writing of this paper. In particular, LLMs were used to check grammar, fix typos and improve readability for the manuscript. No LLMs were used for generating ideas and designing methods.

## A.2 TRAINING TEMPLATE

Our training template is shown in Table 9.

Table 4: PPR's RL training template. `question` will be replaced with the specific query from selected datasets during training and inference.

---

Answer the given question. You must conduct reasoning inside `<think>` and `</think>` first every time you get new information. After reasoning, if you find you lack some knowledge, you can call a search engine by `<search>` query `</search>`, and it will return the top searched results between `<information>` and `</information>`. You can search as many times as you want. If you find no further external knowledge needed, you can directly provide the answer inside `<answer>` and `</answer>` without detailed illustrations. For example, `<answer>` Beijing `</answer>`. Question: `question`.

---

## A.3 TRAINING RECIPE

Following Jin et al. (2025), we set the learning ratess to 1e-6 for 1e-6 for 500 steps with warm-up ratios of 0.285 and 0.015We adopt GAE with $\lambda = 1$ and $\gamma = 1$. The train batch size is set to 512, with mini-batch set to 256 and micro-batch set to 64. Training is performed on a single node with 8 GPUs, with a maximum sequence length of 4,096 tokens, response length of 500, and retrieved content length of 500. Rollouts are generated with vLLM with tensor parallel of 1 and GPU memory ratio of 0.6, sampling with temperature 1.0 and top-p 1.0. The clip ratio is set to $\epsilon = 0.001$ and the KL regularization coefficient $\beta = 0.001$ for 3B models and 0 for 7B models. Models are evaluated and checkpoints saved every 50 steps, with the best checkpoint chosen on validation set. Max turn is set to 2 for General QA and 3 for Multi-Hop QA, with the top-3 passages retrieved by default.

## A.4 DETAILS OF NVPROCESSBENCH CONSTRUCTION

For collecting reliable benchmark data, we adopt the following generation and filtering pipeline:

**Data collection and initial filtering.** We first collect 60k trajectories from the NQ and HotpotQA datasets during RL training of Qwen2.5 models (3B/7B, Base/Instruct) with EM as outcome rewards only. GPT-5 is then used to filter low-quality samples, including unanswerable queries, incorrect gold answers, and extreme poor responses. This initial filtering results in 48k retained trajectories.

**Label annotation.** Each intermediate step is scored according to our defined principles (Appendix A.7) by GPT-4o and Qwen3-235B-A22B. To ensure annotation reliability, we remove anomalies, about 15% of the data, where the score difference between the two judges exceeds 0.4. For the remaining cases with smaller disagreements, we perform manual review and correction in a verification loop to prevent annotation drift.

**Distribution balancing.** The mean step score across the dataset is 0.652, which we use as the threshold for binary labeling: steps with scores above this value are labeled as 1 (good) and those below as 0 (bad). To balance score distributions, we randomly sample 2,000 trajectories while ensuring a balanced combination of process and outcome scores. Specifically, this sample includes (1) 504 "final correct answer + good intermediate steps", (2) 496 "final correct answer + bad intermediate steps", (3) 466 "final incorrect answer + good intermediate steps", (4) 534 "final incorrect answer + bad intermediate steps". During selection, all data undergo a final expert review, with low-quality cases discarded and replaced by new samples from the pool. The final benchmark dataset is balanced in score distribution and consists of 564 multi-step and 1,436 single-step trajectories.

## A.5 DETAILS OF PPRM TRAINING

**Dataset construction.** Following Sec. A.4, we collect an additional 2k trajectories from training on the NQ dataset, ensuring no overlap with the benchmark data, and annotate them using the same dual-model pipeline. This yields an SFT training corpus, which we use to fine-tune Qwen3-8B and obtain our PRR-PRM.

**Training Recipe.** We perform supervised fine-tuning on the Qwen3-8B model for 3 epochs, using a learning rate of 2e5 with cosine decay and a warm-up ratio of 0.1. The training is carried out with a batch size of 32 on a single node with 4 GPUs.

## A.6   CASE STUDIES

To gain deeper insights into PPR, we conduct case studies that examine both the training model's behavior and the process reward model's evaluations. The studies are shown in the following tables.

Table 5: Case Study 1: The process reward model can provide principled evaluations and generate consistent final scores.

---

**Question:** Who was the former co-driver and two-time world champion with the Finnish head of the Toyota GAZOO Racing team?

**Ground Truth:** Risto Mannisenmäki

---

**MODEL:**

`<think>`

To find the person who fits the criteria, break down the query into subparts and search step by step. I need to identify the Finnish head of the Toyota GAZOO Racing team first, as this can lead to the co-driver information.

`</think>`

`<search>` Finnish head of Toyota GAZOO Racing team for co-driver position `</search>`

`<information>`

Doc 1(Title: "Toyota Gazoo Racing WRT") Toyota Gazoo Racing WRT Toyota Gazoo Racing WRT is a World Rally Championship (WRC) team based in Finland that serves as Toyota's official factory team. Its team principal is four time World Drivers' Champion Tommi Mäkinen. The team made its début during the 2017 season, where it entered the Toyota Yaris WRC . . .

`</information>`

> **Process Reward:**
> Analysis: The response correctly identifies the Finnish head of the Toyota GAZOO Racing team as Tommi Mäkinen based on the provided information. It correctly extracts relevant details about Mäkinen's role and achievements. However, the response does not fully address the co-driver and two-time world champion aspect, which is crucial to the query. Additionally, the search query could be more specific to better target the required information.
> Scores: `<final_score>`4,6`</final_score>`

`<think>`

From the information provided, I can identify that the Finnish head of the Toyota GAZOO Racing team is Tommi Mäkinen. Now, I need to look into his co-drivers to find the one who was a two-time world champion.

Now, I need to look for the co-drivers associated with Tommi Mäkinen and check if any of them are two-time world champions

`</think>`

`<search>` Tommi Mäkinen co-drivers two time world champion `</search>`

`<information>`

Doc 3(Title: "Risto Mannisenmäki") Risto Mannisenmäki Risto Mannisenmäki (born on 28 May 1959) is a former rally co-driver and two-times world champion with driver Tommi Mäkinen. Mannisenmäki begun his career in 1982 and was co-driving for various drivers such as Tommi Mäkinen, Sebastian Lindholm and Ari Mökkönen in local rallies. . . .

`</information>`

> **Process Reward:**
> Analysis: The response correctly identifies Tommi Mäkinen as the Finnish head of the Toyota GAZOO Racing team and correctly searches for his co-drivers. It also correctly identifies Risto Mannisenmäki as the former co-driver and two-time world champion with Tommi Mäkinen. All strings are wrapped in the required tags.
> Scores: `<final_score>`6,6`</final_score>`

`<answer>` Risto Mannisenmäki `</answer>`

---

Table 6: Case Study 2: the process reward model can detect flawed reasoning steps even when the final answer is correct.

---

**Question:** Which musician is from Europe, King Diamond or Bob Marley?
**Ground Truth:** King Diamond

---

**MODEL:**
`<think>`
First, I need to determine the origins of both musicians. King Diamond is a German-born American heavy metal singer. Bob Marley is an iconic Jamaican singer and musician. Knowing this, I can reason that one of these musicians is from Europe (Germany) and the other is from the Americas (Jamaica).
`</think>`
`<search>` query `</search>`
`<information>`
Doc 1(Title: "Query (Quaker)") spiritual challenge to their local community of Friends. Examples of Queries: Query (Quaker) Quakers use the term Query to refer to a question or series of questions used for reflection and in spiritual exercises. Friends have used Queries as tools for offering spiritual challenges to the community for much of their history ...
`</information>`

**Process Reward:**
Analysis: The response correctly identifies the origins of King Diamond and Bob Marley, reasoning that one is from Europe (Germany) and the other from the Americas (Jamaica). However, the search query provided is incomplete ("query") and does not reflect the actual question or required information. Additionally, the information provided from the documents is irrelevant to the query, as it discusses topics like "Query (Quaker)", "Query flooding", and "Query string", which do not pertain to the musicians' origins. The search query is not properly aligned with the task.
Scores: `<final_score>` 1,6 `</final_score>`

`<think>`
Since the search results seem to be unrelated to our initial query and I didn't find direct information about the origins of King Diamond and Bob Marley, I need to manually check the information needed.
From `<information>`, I found that King Diamond is indeed a German-born American heavy metal singer, indicating he is from Europe (Germany). Bob Marley is from Jamaica, which is a Caribbean island and part of the Americas.
Thus, the musician from Europe is King Diamond.
`</think>`

`<answer>` King Diamond `</answer>`

---

Table 7: Case Study 3: the process reward model assigns high scores to steps that exhibit correct reasoning and tool usage in accordance with the designed principles.

---

**Question:** Drachen Fire was a roller coaster located behind which suspended coaster at Busch Gardens Williamsburg?
**Ground Truth:** Big Bad Wolf

---

MODEL:
`<think>`
To find the answer to this question, I need to identify the Drachen Fire roller coaster and determine which roller coaster it was situated behind in Busch Gardens Williamsburg. Let's start with the search.
`</think>`
`<search>` Drachen Fire roller coaster Busch Gardens Williamsburg `</search>`
`<information>`
Doc 1(Title: "Drachen Fire") Drachen Fire Drachen Fire was a steel roller coaster located at Busch Gardens Williamsburg, in Williamsburg, Virginia. Operating from 1992 until 1998, the roller coaster was manufactured by Arrow Dynamics. It featured electric-blue track and silver supports, and was located in the Oktoberfest portion of the park, behind the Big Bad Wolf and Das Festhaus. The tagline for the ride was "Feel the Heat." The ride had three separate trains, featuring seven cars each, with each car holding passengers arranged two by two...
`</information>`

**Process Reward:**
Analysis: The response correctly extracts information from the provided documents, identifying that Drachen Fire was located behind the Big Bad Wolf and Das Festhaus at Busch Gardens Williamsburg. The search query used is appropriate for finding information about the Drachen Fire roller coaster. The response correctly decides to conduct a search to find the specific information needed to answer the question.
Scores: `<final_score>` 3,3 `</final_score>`

`<search>` Big Bad Wolf Das Festhaus Drachen Fire `</search>`
`<information>`
Doc 1(Title: "Drachen Fire") Drachen Fire Drachen Fire was a steel roller coaster located at Busch Gardens Williamsburg, in Williamsburg, Virginia. Operating from 1992 until 1998, the roller coaster was manufactured by Arrow Dynamics. It featured electric-blue track and silver supports, and was located in the Oktoberfest portion of the park, behind the Big Bad Wolf and Das Festhaus. The tagline for the ride was "Feel the Heat." ...
`</information>`

**Process Reward:**
Analysis: The response correctly extracts information from the provided documents, identifying that Drachen Fire was located behind the Big Bad Wolf and Das Festhaus. It also provides a relevant search query to further confirm this information. All strings are wrapped in the appropriate tags.
Scores: `<final_score>` 3,3 `</final_score>`

Based on the information gathered:
Drachen Fire was a steel roller coaster located at Busch Gardens Williamsburg. It was situated in the Oktoberfest portion of the park, behind the Big Bad Wolf and Das Festhaus. Therefore, the roller coaster situated behind Drachen Fire at Busch Gardens Williamsburg was Big Bad Wolf.
`<answer>` Big Bad Wolf `</answer>`

---

Table 8: Case Study 4: the process reward model can evaluate the quality of search queries based on retrieved information and prior steps, identifying issues such as imprecise search queries or redundant search actions.

---

**Question:** What type of category does Echium and Oxydendrum have in common?
**Ground Truth:** genus

---

**MODEL:**
`<think>`
To determine the category that Echium and Oxydendrum have in common, I'll start by gathering information on these plants. First, let's think about what kinds of plants these might be. They both sound botanical and their names suggest they might be related to family or genus classifications. Let's search to learn more about them.
`</think>`
`<search>` Echium and Oxydendrum classification `</search>`
`<information>`
Doc 1(Title: Echinodorus) Echinodorus Echinodorus, the burhead or Amazon sword plant, is a genus of plants in the family Alismataceae and distributed in the Western Hemisphere from the central United States to Argentina. Its scientific name is derived from Ancient Greek "echius" - rough husk, and "doros" - leathern bottle . . . Doc 2(Title: Oxybasis) Oxybasis Oxybasis is a flowering plant genus from the subfamily Chenopodioideae in the Amaranthaceae family . . .
`</information>`

**Process Reward:**
Analysis: The response correctly identifies the need to gather information on Echium and Oxydendrum and formulates a relevant search query. However, the provided `<information>` does not contain data on Echium or Oxydendrum, so no useful information is extracted. The response does not make a decision on whether to conduct further search, but since the initial search did not yield results, it would be appropriate to consider alternative search terms or sources. The response adheres to formatting rules.
Scores: `<final_score>` 2,6 `</final_score>`

`<search>` Echium and Oxydendrum classification `</search>`
`<information>`
Doc 1(Title: Echinodorus) Echinodorus Echinodorus, the burhead or Amazon sword plant, is a genus of plants in the family Alismataceae and distributed in the Western Hemisphere from the central United States to Argentina. Its scientific name is derived from Ancient Greek "echius" - rough husk, and "doros" - leathern bottle . . . Doc 2(Title: Oxybasis) Oxybasis Oxybasis is a flowering plant genus from the subfamily Chenopodioideae in the Amaranthaceae family . . .
`</information>`

**Process Reward:**
Analysis: The response repeats the same search query and information provided in the context without adding any new analysis or synthesis. It does not attempt to extract relevant information about the classification of Echium and Oxydendrum, nor does it provide a correct search query based on the information given. Additionally, it fails to decide whether to conduct a search or if the existing information is sufficient.
Scores: `<final_score>` 0,6 `</final_score>`

## A.7 PROMPT OF PROCESS REWARD MODEL

Table 9: Prompt design of process reward model.

**Process Reward Model Template**

**System Prompt:**

You are a very strict and skilled evaluator.
Given **Query** and **Response** pair, you should only evaluate the **Response**. This **Response** is one turn of multi-turn response, so it is acceptable that the response does not have final answer.
For evaluation, generate reasonable principles and score each principle individually.
**The most important principles you can refer are:**

1. Whether it correctly extracts information from `<information>` based on the query.

2. Whether it provides correct search query for `<search>`.

3. Whether it correctly decides conduct or not conduct `<search>`.

Every string in **Response** must be wrapped in `<think></think>`, `<search></search>`, `<information></information>`, or `<answer></answer>`. Otherwise, set the output SCORE to 0.

**[Output Format Requirements]**
Respond in exactly two lines:
1. Analysis: Explain the reasoning and individual scores for each principle.
2. Scores: `<final_score>`SCORE,MAX_SCORE`</final_score>`. e.g., `<final_score>`4,6`</final_score>`.

----------------------------------------------------------------

**User Prompt:**

**[Conversation Context]**
**Query**: *query_str*
**Response**: *solution_str*

A.8    THEORETICAL ANALYSIS FOR RENORM

As describe in Section 3.2, we further discuss why centralizing process rewards with outcome rewards proposed in ReNorm is effective as follows:

Let discrete $Y := r_o \in \{0, 1\}$ and continuous variant $X := r_{pt} \in [0, 1]$ for a fixed turn $t$. Then:

[Boundedness and class-conditional separation]

We have $\tilde{r}_t = X + Y - 1$ satisfies the bounded range above, and

$$\mathbb{E}[\tilde{r}_t \mid Y = 1] - \mathbb{E}[\tilde{r}_t \mid Y = 0] \;=\; \big(\mathbb{E}[X \mid Y = 1] - \mathbb{E}[X \mid Y = 0]\big) + 1 \;\geq\; 1,$$

with equality iff $\mathbb{E}[X \mid Y = 1] = \mathbb{E}[X \mid Y = 0]$. Thus broadcasting $Y$ introduces at least a unit margin between the correct/incorrect class means, strictly improving discriminability whenever the PRM is even weakly informative ($\mathbb{E}[X \mid Y = 1] > \mathbb{E}[X \mid Y = 0]$).

*Proof.* The range follows from $X \in [0, 1]$ and $Y \in \{0, 1\}$. The mean gap is $\big(\mathbb{E}[X \mid 1] + 1 - 1\big) - \big(\mathbb{E}[X \mid 0] + 0 - 1\big) = (\mathbb{E}[X \mid 1] - \mathbb{E}[X \mid 0]) + 1$. □

[Unbiasedness under symmetric priors; general centering]

If $\mathbb{E}[X] = \mathbb{E}[Y] = \frac{1}{2}$, then $\mathbb{E}[\tilde{r}_t] = 0$. In general,

$$\mathbb{E}[\tilde{r}_t] = \mu + \pi - 1, \qquad \mathbb{E}[\bar{r}_t] = 0.$$

Hence subtracting the constant $\mu + \pi - 1$ centers $\tilde{r}_t$ without altering its variance.

*Proof.* Linearity of expectation. □

[Variance decomposition and stability]

Let $\mathrm{Var}(X) = \sigma_X^2$, $\mathrm{Var}(Y) = \pi(1 - \pi)$, and $\mathrm{Cov}(X, Y) = \rho \, \sigma_X \sqrt{\pi(1 - \pi)}$ with correlation $\rho \in [-1, 1]$. Then

$$\mathrm{Var}(\tilde{r}_t) \;=\; \sigma_X^2 + \pi(1 - \pi) + 2\,\mathrm{Cov}(X, Y) \;\leq\; \sigma_X^2 + \tfrac{1}{4} + 2|\rho|\,\sigma_X/2.$$

When $X$ is positively correlated with correctness ($\rho > 0$), broadcasting $Y$ increases signal-to-noise in the *class-conditional* sense (Prop. A.8); centering by a constant (either 1 or $\mu + \pi$) does not change variance but reduces the magnitude of bootstrapped TD-errors $\delta_t = \tilde{r}_t + \gamma v_{t+1} - v_t$, yielding lower-variance advantages $A_t$ under GAE.

*Proof.* Apply $\mathrm{Var}(A + B) = \mathrm{Var}(A) + \mathrm{Var}(B) + 2\mathrm{Cov}(A, B)$ and the Bernoulli variance bound $\pi(1 - \pi) \leq 1/4$. Centering subtracts a constant, leaving variance unchanged. □

[Objective alignment]

With generalized centering $\bar{r}_t = (X - \mu) + (Y - \pi)$, the expected policy objective $\mathbb{E}[\sum_t \bar{r}_t]$ is proportional to $\sum_t \mathrm{Cov}(X_t, Y)$. If the PRM is *monotone informative* in the sense that improving step quality increases both $X_t$ and the success probability $\mathbb{P}(Y = 1)$, then $\mathrm{Cov}(X_t, Y) > 0$ and gradient ascent on $\bar{r}_t$ aligns with improving end-task accuracy.

*Proof.* $\mathbb{E}[\bar{r}_t] = (\mathbb{E}[XY] - \mu\pi) = \mathrm{Cov}(X, Y)$ by bilinearity and centering. Monotone informativeness implies positive covariance. □

### A.9 REBUTTAL

#### A.9.1 CONVEX COMBINATION WITH $r_p$ AND $r_o$

We evaluate a convex combination baseline defined as

$$r_{p,t} = 2 \times (m \cdot \hat{r}_{p,t} + (1 - m) \cdot r_o) - 1,$$

with $m = 0.25$ and $m = 0.75$. We conduct the evaluations on Qwen2.5-3B-Base across single-hop QA benchmarks, as shown below:

Table 10: Convex Combination

| Method | NQ | TriviaQA | PopQA | Avg. |
|---|---|---|---|---|
| $m = 0.25$ | 0.219 | 0.315 | 0.216 | 0.250 |
| $m = 0.75$ | 0.343 | 0.501 | 0.390 | 0.410 |
| **Ours (ReNorm)** | 0.423 | 0.565 | 0.411 | 0.463 |

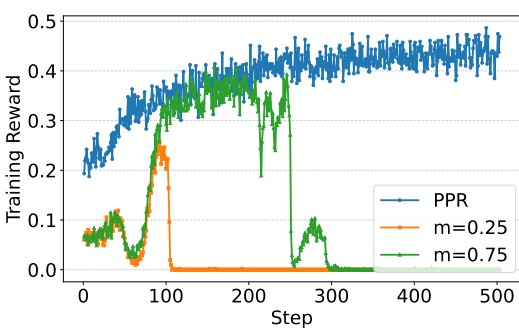

Figure 6: Convex Combination Baselines v.s. ReNorm

The results and Figure 6 show that our approach consistently outperforms the convex combination baseline across these tasks without collapse.

#### A.9.2 WITH V.S. WITHOUT SHIFT

To further demonstrate the effectiveness of our method, we provide the evaluation of ReNorm with and without the constant shift. The baseline without shift means we simply use $r_p + r_o$ instead of Eq. 7 as the process reward. Simply use $r_p$ as the process reward has already been discussed in Table 3. The following are the results evaluated with Qwen2.5-3B-Base on single-hop QA benchmarks.

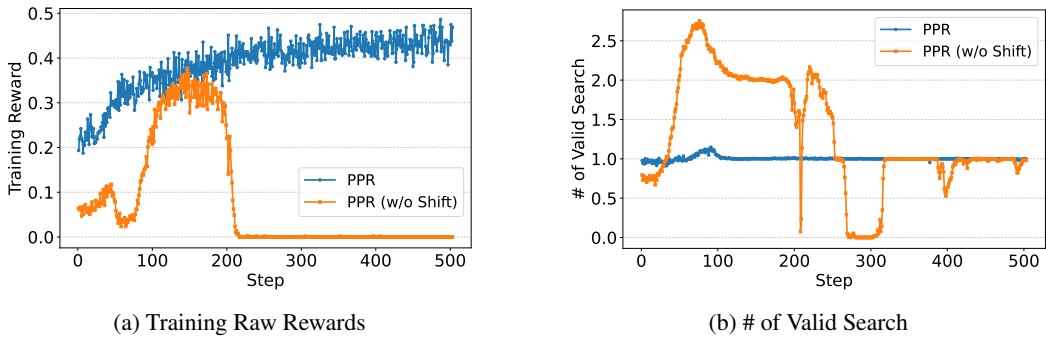

(a) Training Raw Rewards          (b) # of Valid Search

Figure 7: ReNorm v.s. Baselines withour shift

We observe the reward collapse occurs around 200 steps for the baseline (Figure 7a), and the number of valid search has incrementally increased to the maximum search attempts we set during the initial 100 steps. Then it fell down with fluctuation (Figure 7b). In contrast, PPR with ReNorm retains stable training rewards and converge to the best # of search attempts (1) during training.

### A.9.3 DYNAMIC TUNING OF POSITIVE AND NEGATIVE RATIO

We implemented a dynamic ratio by subtracting batch-wise means of both process and outcome rewards for a single rollout batch size: $R = (r_p - r_{batch,p}) + (r_o - r_{batch,o})$. We evaluate this dynamic ratio strategy on Qwen2.5-3B-Base using the same benchmarks as mentioned before:

Table 11: Dynamic Tuning Baseline v.s. ReNorm

| Method | NQ | TriviaQA | PopQA | Avg. |
|---|---|---|---|---|
| Dynamic | 0.406 | 0.546 | 0.433 | 0.462 |
| **Ours** | 0.423 | 0.565 | 0.411 | 0.463 |

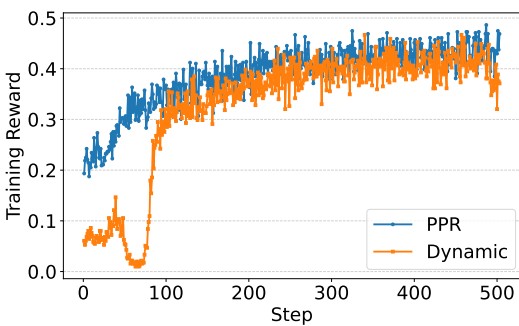

Figure 8: Dynamic Tuning Baseline v.s. ReNorm

These results and Figure 8 suggest that dynamically tuning the positive–negative ratio is not a primary factor influencing our RL training. The baselines performs reasonably but still consistently underperforms ReNorm. We attribute this to early-stage instability: the batch-wise ratio fluctuates substantially at the beginning of RL training and introduce additional variance into the advantage estimates. This observation aligns with the reward curve during the initial 100 steps in the figure.

### A.9.4 MORE METRICS ON STABILITY

We further provide two more metrics on to demonstrate our methods' effectiveness.

Gradient norm reflects the magnitude of policy updates, and entropy loss tracks how quickly the policy collapses toward determinism. Both metrics are the key indicators of training stability. As shown in Figure 9, PPR is the only method that maintains a smooth and bounded gradient norm, while the other normalization strategies exhibit severe gradient explosions around step 350, signaling unstable optimization. The entropy loss curves show the same pattern: PPR produces a stable, gradual decrease, whereas the others display premature entropy collapse or sharp spikes, indicating unstable or overly aggressive updates. Overall, these metrics consistently show that PPR provides the most stable and reliable training dynamics.

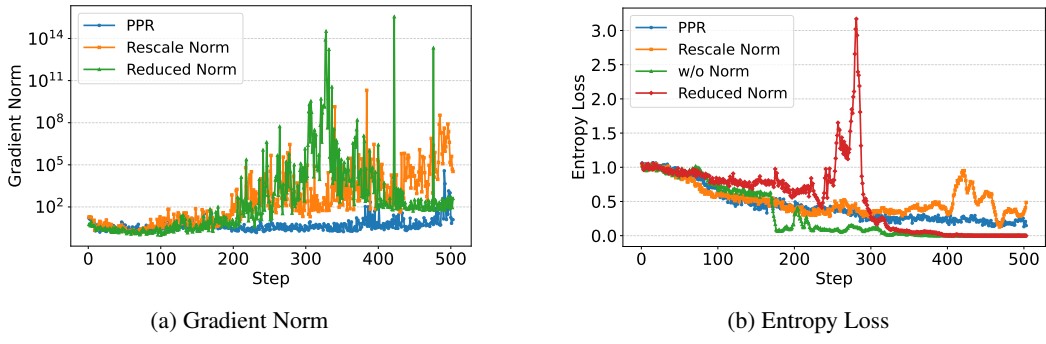

(a) Gradient Norm                    (b) Entropy Loss

Figure 9: PPR v.s. Baselines on gradient norm and entropy loss

### A.9.5 General Theoretical Analysis for ReNorm

Derived from section A.8. We now have no assumptions on $\mu = \pi = \frac{1}{2}$.

Let $X := r_{pt} \in [0, 1]$ denote the process score at turn $t$ and $Y := r_o \in \{0, 1\}$ the outcome reward. Write their means as $\mu := \mathbb{E}[X]$ and $\pi := \mathbb{E}[Y]$, and define the centered variables $\tilde{X} := X - \mu$ and $\tilde{Y} := Y - \pi$. For nonnegative weights $a, b \geq 0$, consider the centered affine fusion

$$\bar{r}_t := a\,\tilde{X} + b\,\tilde{Y} = a\,(X - \mu) + b\,(Y - \pi).$$

This subsumes the unweighted case $a = b = 1$ used in practice.

[Zero mean and bounded range] We have $\mathbb{E}[\bar{r}_t] = 0$ and

$$\bar{r}_t \in \big[-a\mu - b\pi,\ a(1 - \mu) + b(1 - \pi)\big].$$

In particular, for $a = b = 1$, the range is $\big[-(\mu + \pi),\ 2 - (\mu + \pi)\big]$.

*Proof.* $\mathbb{E}[\bar{r}_t] = a\,\mathbb{E}[X - \mu] + b\,\mathbb{E}[Y - \pi] = 0$ by centering. The range follows from $X \in [0, 1]$ and $Y \in \{0, 1\}$. $\square$

[Class-conditional separation] Let $\Delta_X := \mathbb{E}[X \mid Y = 1] - \mathbb{E}[X \mid Y = 0] \in [-1, 1]$. Then

$$\mathbb{E}[\bar{r}_t \mid Y = 1] - \mathbb{E}[\bar{r}_t \mid Y = 0] = a\,\Delta_X + b.$$

Hence for any $b > 0$ there is at least a margin $b$ between correct/incorrect classes, and the margin is strictly larger than $b$ whenever the PRM is informative ($\Delta_X > 0$).

*Proof.* $\mathbb{E}[\tilde{X} \mid Y = i] = \mathbb{E}[X \mid Y = i] - \mu$ and $\mathbb{E}[\tilde{Y} \mid Y = i] = i - \pi$ for $i \in \{0, 1\}$. Subtracting the two conditionals gives the identity. $\square$

[Variance decomposition and stability] Let $\sigma_X^2 := \mathrm{Var}(X)$, $\mathrm{Var}(Y) = \pi(1 - \pi)$, and $\mathrm{Cov}(X, Y) = \rho\,\sigma_X\sqrt{\pi(1 - \pi)}$ with $\rho \in [-1, 1]$. Then

$$\mathrm{Var}(\bar{r}_t) = a^2\sigma_X^2 + b^2\pi(1 - \pi) + 2ab\,\mathrm{Cov}(X, Y) \leq a^2\sigma_X^2 + \frac{b^2}{4} + ab\,|\rho|\,\sigma_X.$$

For $\rho > 0$ (process scores positively correlated with correctness), class-conditional discrimination improves (Prop. A.9.5). Centering by construction leaves the variance term unchanged while reducing the magnitude of bootstrapped TD errors $\delta_t = \bar{r}_t + \gamma v_{t+1} - v_t$, yielding lower-variance advantages under GAE.

*Proof.* Apply $\mathrm{Var}(A + B) = \mathrm{Var}(A) + \mathrm{Var}(B) + 2\mathrm{Cov}(A, B)$, the Bernoulli bound $\pi(1 - \pi) \leq \frac{1}{4}$, and Cauchy–Schwarz for $\mathrm{Cov}$. Centering subtracts constants and does not affect variance. $\square$

[Objective alignment] For any $a, b > 0$,

$$\mathbb{E}[\bar{r}_t \mid Y = 1] - \mathbb{E}[\bar{r}_t \mid Y = 0] = a\,\Delta_X + b.$$

Thus increasing the success probability $\mathbb{P}(Y = 1)$ strictly increases $\mathbb{E}[\bar{r}_t]$ even when $\Delta_X = 0$, and yields a larger gain when the PRM is monotone informative ($\Delta_X > 0$). Consequently, gradient ascent on the return formed by $\bar{r}_t$ is aligned with improving end-task accuracy, while still crediting intermediate quality through $a\,\Delta_X$.

*Proof.* By the law of total expectation, $\mathbb{E}[\bar{r}_t] = \mathbb{P}(Y = 1)\,\mathbb{E}[\bar{r}_t \mid Y = 1] + \mathbb{P}(Y = 0)\,\mathbb{E}[\bar{r}_t \mid Y = 0]$. Differentiating w.r.t. $\mathbb{P}(Y = 1)$ yields the class-conditional gap in Prop. A.9.5. $\square$

