# OpenReview forum: "Principle Process Reward For Search Agents"
_ICLR.cc/2026/Conference — Submitted to ICLR 2026_

### Official Review · Reviewer_ENAr · 2025-10-29

**Soundness:** 3
**Presentation:** 2
**Contribution:** 3
**Rating:** 6
**Confidence:** 2

**Summary:**

The paper proposes Principle Process Reward (PPR), a RL framework for LLM-based search agents that combines outcome and step-level supervision.
It introduces (1) a Principle-based Process Reward Model (PPRM) that evaluates intermediate steps via interpretable principles and (2) a Reward Normalization (ReNorm) method that aligns process and outcome rewards for stable training. Their method is evaluated on multiple QA and multi-hop benchmarks with Qwen2.5 models. PPR achieves up to 28% gains over non-RL baselines and 15% over RL counterparts, showing improved stability and generalization. The authors also released NVProcessBench, a benchmark for non-verifiable process evaluation.


The paper is clear, well-organized, and technically sound. It presents strong empirical results and thoughtful ablations. Its main contribution, combining principle-based process rewards with reward normalization, offers a practical and interpretable improvement to RL-based LLM training. However, the idea builds on prior PRM work. The scope is limited to search-based QA, and limitations are only discussed implicitly.

**Strengths:**

+ The integration of principle-based process evaluation with outcome reward normalization is conceptually clear and empirically effective.

+ Extensive comparisons against baselines show consistent and significant gains across both in-domain and out-of-domain QA tasks.

+ NVProcessBench is a meaningful step toward systematic evaluation of reward models for non-verifiable reasoning steps.

+ The paper is generally well written, with clear motivation, architecture diagrams, and quantitative evidence.

**Weaknesses:**

+ While ReNorm is intuitively and empirically justified, its theoretical analysis in Appendix A.9 is limited to simple assumptions and does not fully explain stability in more general cases.

+ All evaluations are in search-based QA. It remains unclear whether the approach generalizes to other non-verifiable agentic tasks such as code generation, planning, or multi-tool reasoning.

+ PPRM relies on a fine-tuned 8B model; cost and scalability implications are not deeply analyzed.

+ Methods such as reward rescaling, entropy regularization, or variance reduction are briefly mentioned but not compared much.

+ NVProcessBench is built partly using GPT-based annotation, which may introduce bias; details on annotation validation are limited. How do you address this ?

+ The figures are difficult to read due to dense layouts and low-contrast color schemes; improving visual clarity would make the results and framework easier to interpret.

+ The paper acknowledges some limitations. However, it does not explicitly discuss broader constraints, such as limited domain coverage, scalability to other agentic tasks, or computational overhead of PPRM training. Future directions is also given more implicitly.

## Points to Improve

*This is not meant for the rebuttal (see below), but general points that could strengthen the paper*

+ Extend experiments to other domains (e.g., math reasoning, code tasks) to demonstrate generality beyond search.

+ Provide more rigorous theoretical or empirical justification of why ReNorm ensures stability, e.g., via variance or gradient norm analysis during PPO training.

+ Analyze the cost and efficiency of training the PPRM relative to baseline PRMs setups.

+ Clarify bias and reproducibility aspects of NVProcessBench, especially regarding synthetic labels from GPT models.

+ A.1 as written is not accurate. LLMs were also used in benchmark annotation (see A5 and A6), so that usage should be explicitly acknowledged to maintain full transparency and reproducibility compliance. I would not say the current form is unethical or try to hide the use, it is just a little inconsistent.

+ The figures could be improved for clarity and accessibility. Consider using colorblind-friendly palettes instead of red–green contrasts, and ensure that legends are placed outside or below the plots so they don’t overlap with the lines. Improving visual contrast and spacing would make the results easier to read. Further, each caption should make its figure a standalone component of the paper. Thus, each figure should be built in the following way:
* The first sentence should highlight the main message of the Figure/Table (e.g. "Our method outperforms the existing SOTA methods on the studied problem.")
* The next sentences then explain what is depicted in the Table/Figure. E.g. Mean test accuracy, on 5 seeded trainings, with std.
* Finally, details and references to e.g. appendix can be provided if necessary. E.g. Our method outperforms baseline 1 in 3 out of 4 tasks, ... etc.

In your captions, the answer to the addressed research question is missing. For example, in Figure 3: The joint used of all components included in PPR allow it to provide more stable reward than the compared baselines... (could be better formulated).
Put this sentence in bold.

Nit: Typos:
In abstract: Experiment*al* results

**Questions:**

1. Did you make a small ablation about how PPR performs in non-search domains? Do you have reasons to believe that it would not work with other types of tools ?

2. Can the authors provide empirical evidence that ReNorm reduces gradient variance compared to other normalization schemes?

3. How large is the training cost overhead (compute and/or wall-clock time) relative to baseline RLVR methods?

4. Does NVProcessBench include any human validation to confirm that GPT-based step labels align with human judgments?

---

> ### Author Response · Authors · 2025-11-20
>
> We appreciate your valuable feedback. We will incorporate the clarifications and address the issues in the next draft. Here is our response to your questions:
>
> ---
>
> > its theoretical analysis in Appendix A.9 is limited to simple assumptions and does not fully explain stability in more general cases.
>
> Thanks for pointing out. We provide an extend version of the Appendix A.9.5 in the latest version.
>
> > PPRM relies on a fine-tuned 8B model; cost and scalability implications are not deeply analyzed. And the training cost compared to RLVR baseline need to be discussed.
>
> We highlight the discussion of PPRM in Section 3.3. In practice, PPRM is lightweight: we fine-tuned an 8B model with ~2k trajectories in 2 hours. PPRM can achieve performance comparable to, or even better than larger judge models up to 235B or reasoning models (ThinkPRM). While Large models as reward models are typically expensive to obtain or deploy for high API costs and serving compute, our lightweight SFT produces a high-quality reward model at a small fraction of the cost.
>
> Regarding training cost relative to the RLVR baseline, integrating PPRM introduces limited overhead. During RL, we deploy PPRM on 4 GPUs via SGLang. In the Qwen2.5-3B setting, finishing a batch of process rewards computation takes less 10 seconds per step, resulting in less than one hour of additional time compared with the ORM-only baselines, which itself requires roughly 15 hours. We consider this overhead minor and practically acceptable, especially given the training-stability and performance gains brought by PPRM.
>
> > Methods such as reward rescaling, entropy regularization, or variance reduction are briefly mentioned but not compared much.
>
> We agree that normalization strategies are crucial and deserve comprehensive comparison and analysis. We have already evaluated multiple normalization variants in Table 3. Additionally, following Reviewer 1’s suggestion, we have included experiments using a convex combination of $r_p$ and $r_o$, further broadening the comparison.
>
> > details on annotation validation for NVProcessBench are limited
>
> We first use a multi-model scoring setup: multiple models independently score each process sample, and we filter out instances where the votes remain highly inconsistent across multiple attempts, which reduces the bias introduced by using any single model as a judge. We then perform additional reviews to further remove potential low-quality cases and mitigate residual bias. We will include these details on this annotation and validation procedure in the future version.
>
> > Extend experiments to other domains (e.g., math reasoning, code tasks) to demonstrate generality beyond search
>
> This paper focuses specifically on search-based agent tasks, where non-verifiable process rewards are particularly challenging and central to the problem. We agree that evaluating our method on additional domains would further strengthen its generality. From a methodological standpoint, our pipeline is modular and can be easily adapted to other tasks by adjusting the corresponding principles, and we are currently conducting experiments on code-generation task to verify this. We will update the results as soon as they are available.
>
> > Provide more rigorous theoretical or empirical justification of why ReNorm ensures stability, e.g., via variance or gradient norm analysis during PPO training
>
> We have added the gradient-norm and entropy-loss metrics in the revised PDF (see Appendix A.9.4 for details). Empirically, ReNorm markedly reduces entropy fluctuations and stabilizes gradient norms, preventing gradient explosion/vanishing and avoiding collapse.
>
> Theoretically, our step-level process rewards naturally produces a smoother reward landscape in range [0,1], instead of being concentrated at two discrete points for outcome reward. This reduces the per-trajectory reward gap across samples, thereby lowering the variance of the advantage estimator and leading to more stable policy-gradient updates.
>
> > Writing suggesstions
>
> We thank the reviewers for the detailed writing suggestions. We will incorporate improved figure color schemes and the clarified LLM statement in the future version.

---

### Official Review · Reviewer_FRDx · 2025-11-04

**Soundness:** 3
**Presentation:** 3
**Contribution:** 2
**Rating:** 2
**Confidence:** 5

**Summary:**

This paper proposes a reinforcement learning framework for training search agents, integrating both outcome rewards and process rewards. Additionally, the authors designed and trained a generative process reward model and further introduced a reward normalization strategy to combine outcome and process rewards. The framework was evaluated on multiple datasets. The authors also provided a new benchmark to assess scenarios where process rewards are non-quantifiable.

**Strengths:**

1. This paper models the process reward, thereby enriching the training pipeline for the search agent.
2. The authors introduce a new benchmark to evaluate scenarios where process rewards are non-verifiable.
3. The experimental design  is well-executed.

**Weaknesses:**

1. The proposed reward normalization method seems to deviate from the standard understanding of PPO and GAE. Reward whitening is a very common trick in PPO training. However, in the proposed "ReNorm" method, the authors simplistically transform it into subtracting 1/2 from both the outcome and process rewards. Essentially, subtracting a \textbf{constant} from the rewards \textbf{has ​no impact​ on} the calculation of GAE. Therefore, the motivation behind this approach is fundamentally flawed.

2. Furthermore, it is rather rash to directly assume a 1:1 ratio between positive and negative advantage samples. Empirical observation shows that the proportion of positive to negative advantages is \textbf{not balanced} during the training process.

3. This paper directly employs a GenRM to output a numerator and denominator for non-verifiable process rewards, but it lacks a description of the golden-answer verification for the outcome reward. In reality, especially in search scenarios, the rewards for an agent's search process are often quantifiable (e.g., whether a tool call was successful, whether the returned text is highly relevant to the query). However, a quantifiable metric for the final answer produced by the agent is often missing.

Based on the above three points, the motivation and contributions of this paper are flawed.

**Questions:**

Same as Weaknesses.
1. Subtracting a \textbf{constant} from the rewards \textbf{has ​no impact​ on} the calculation of GAE.
2. Empirical observation shows that the proportion of positive to negative advantages is \textbf{not balanced} during the training process.
3. In search scenarios, outcome rewards are often more difficult to quantify than process rewards.

---

> ### Author Response · Authors · 2025-11-20
>
> We appreciate your valuable feedback. We will incorporate the clarifications and address the issues in the next draft. Here is our response to your questions:
>
> ---
>
> > subtracting a constant from the rewards has no impact on the calculation of GAE.
>
> We first clarify, from a mathematical standpoint, why subtracting a constant from all outcome and process rewards **does change** the GAE calculation.
> Let GAE be defined as:
>
> $$A_t = \sum_{l=0}^{T-t-1} (\gamma \lambda)^l \delta_{t+l}$$
> and the TD residual:
>  $$δ_t=r_t+γV(s_{t+1})−V(s_{t})$$
>
> where $r_t$ is the reward and $V$ is the value function, $\gamma$ is the discount factor and $\lambda$ is the trace decay GAE parameter. If we shift reward by a constant $c$, and plug it into GAE, we get:
> $A_t'=\sum_{l=0}^{T-t-1} (\gamma\lambda)^l (\delta_{t+l} - c) = A_t-c\sum_{l=0}^{T-t-1} (\gamma\lambda)^l = A_t - cF(t)$
> where $F(t)=∑_{l=0}^{T−t−1}(γλ)^l$ is a **time-dependent** term. Because the shift depends on the remaining horizon $T-t$, it **cannot** be absorbed by the value function $V(s_t)$, which is a function of the **state** rather than the timestep. In our setting, process rewards are emitted at **turns**-wise. Since different trajectories have different number of turns, this time-dependent shift directly distorts the advantage estimates and consequently the policy gradients. This is the property of applying stepwise process reward at each turn compared with ORM-based only method.
>
> We emphasize this effect by providing additional empirical results (Appendix A.9.2), where the comparison shows that applying this constant adjustment improves training accuracy by **23%** on the NQ dataset (see Appendix A.9.2 for details). Without subtracting the constant, the model learns to **hacking rewards by increasing the number of turns** to accumulate more reward. This unintended behavior mislead the model's behavior thus making the policy shifting from the correct optimizing target. In contrast, when we subtract the constant, the model learns to solve the problems with a reasonable number of turns and avoid unstable training.
>
> Both the theoretical and empirical results indicate the constant reward shift is necessary in our multi-turn RL recipe.
>
> >  it is rather rash to directly assume a 1:1 ratio between positive and negative advantage samples
>
> We agree that the positive and negative sample ratio naturally varies across datasets and throughout RL training in reality. Our choice of a 1:1 ratio is motivated by **stability and robustness** across different datasets and training stages, providing a general-purpose design to avoid distribution-dependent fluctuations. To illustrate this, we implemented a **dynamic ratio** by subtracting batch-wise means of both process and outcome rewards in training: $R=(r_p-r_{batch,p})+(r_o-r_{batch,o})$
>
> As shown in the table below, this dynamic variant consistently underperforms our 1:1 setting. We attribute this to early-stage instability: the batch-wise ratio fluctuates substantially at the beginning of RL training, introducing additional variance into the advantage estimates, which aligns with the observed drop in training rewards when using the dynamic ratio (see Appendix A.9.2 for details).
>
> | Method   | NQ    | TriviaQA | PopQA |
> |----------|-------|----------|-------|
> | Dynamic  | 0.406 | 0.546    | 0.433 |
> | **Ours** | **0.423** | **0.565**    | **0.411** |
>
> Therefore, we find that the **fixed 1:1 ratio is easier to implement and ensure stable and reliable training**.
>
>
> >  it lacks a description of the golden-answer verification for the outcome reward
>
> In this work, we focus on the process reward design for search agents, so we keep the outcome reward aligned with prior work [1,2] by using verifiable QA benchmarks. This setting ensures comparability to prior studies.
>
> The core contributions of our work are introducing principle process reward model for non-verifiable process and proposing ReNorm for unifying process/outcome rewards, which yields clear improvements over ORM-only baselines. While we agree that many real tasks involve non-verifiable outcome rewards, it is beyond the scope of this paper and we leave them for future works.
>
> >  the rewards for an agent's search process are often quantifiable
>
> While tool-use actions such as search calls are observable, the rewards associated with an agent’s search process are not inherently quantifiable. A successful tool invocation does not necessarily indicate progress toward the correct answer—agents may simply learn to call the search engine more frequently to hack the positive reward, regardless of whether the retrieved content is relevant and have contribution towards the final answer. These intermediate steps are therefore not naturally quantifiable, which is precisely why we introduce a principled reward model to more comprehensively evaluate and assign reasonable credits.
>
> [1] Jin, et al, 2025, https://arxiv.org/abs/2503.09516
>
> [2] Li, et al, 2025, https://arxiv.org/abs/2501.05366

---

> ### Comment · Reviewer_FRDx · 2025-11-20
> **Reward shifting in GAE calculation produces no change in results. And this undermines the credibility of the experimental results in this paper.**
>
> In PPO, subtracting a constant  c  from all rewards does not affect GAE calculations. The key reason is that the value network V(s) automatically adjusts by a uniform constant offset \frac{c}{1-\gamma} (where \gamma is the discount factor), as both the Q-function and value function shift consistently with the reward reduction.
> This offset cancels out in the TD-error formula \delta_t = r_t + \gamma V(s_{t+1}) - V(s_t) : the constant term from the reward adjustment is eliminated by the synchronized shift of the value network, leaving the TD-error unchanged. Since GAE is a weighted sum of TD-errors, its result remains unaffected.
>
> In your derivation, why does the value of  V (value function) remain unchanged when only the value of  r (reward) is altered? In practice, V is the expectation of Q (Q-function); if r changes, Q will also change, and consequently V will change as well. You should carefully review your derivation.
>
>
> My primary concern regarding this work lies in the issue of reward shift. Given that such a shift does not cause any changes to GAE, it is necessary to analyze why the experiments yield such significant performance improvements. Additionally, are there any other engineering tricks that are affected by this reward shift?

---

> ### Comment · Reviewer_FRDx · 2025-11-21
>
> Even within the GRPO framework, a mere shift in the reward function is incapable of affecting the advantage calculation. More fundamentally, as established by Definition 1 in [1, 2], any action-independent transformation of the reward function, which includes but is not limited to a constant shift, leaves the optimal policy unchanged. If the primary contribution of this paper is indeed ReNorm, then the authors should check their entire workflow, as the claimed effects may be predicated on a flawed premise.
>
> [1] Direct Preference Optimization: Your Language Model is Secretly a Reward Model https://arxiv.org/abs/2305.18290
>
> [2] From r to Q∗: Your Language Model is Secretly a Q-Function https://arxiv.org/abs/2404.12358

---

> > ### Author Response · Authors · 2025-11-21
> >
> > 1. We would like to clarify a key point from our earlier statement: **the advantage is shifted by a time-dependent term $F(t)$, not a constant offset, the value function $V(s_t)$—which is a function of state, not timestep—cannot fully absorb this shift.** In other words, while the value function can certainly change and adapt to this shift, it cannot compensate for this time-varying shift synchronously, especially when new function-call turns introduce additional turn-level rewards. We also emphasize that our reward design operates at the token level with assigned step rewards, rather than relying solely on a single scalar outcome score. This is a fundamental distinction from outcome-only reward formulations such as DPO.
> >
> > 2. Our work is motivated by a principled framework for designing process rewards for non-verifiable steps and a unified strategy that integrates process and outcome rewards through token-level reward assignment. This is supported by extensive experiments validating the effectiveness of our methodology. Our contributions and empirical performance remain clear and strong.

---

> ### Comment · Reviewer_FRDx · 2025-11-21
>
> Here is a detailed step-by-step derivation.
> 1. reward shift:  $r' = r - c $
> 2. Q syncs shift : $ Q' = \mathbb{E}[\sum \gamma^k r'] = Q - \frac{c}{1-\gamma} $ (summation of a geometric progression with gamma < 1)
> 3. V syncs shift: $ V' = \mathbb{E}[Q'] = V - \frac{c}{1-\gamma} $
> 4. TD-error invariant: $ \delta' = r' + \gamma V'(s_{t+1}) - V'(s_t) = r + \gamma V(s_{t+1}) - V(s_t) = \delta  $
> 5. GAE1: $ \hat{A} = \sum_{k=0}^{\infty} (\gamma\lambda)^k \delta_{t+k} $
> 6. GAE2: $ \hat{A}' = \sum_{k=0}^{\infty} (\gamma\lambda)^k \delta'_{t+k} $
> 7. $\hat{A}' = \hat{A}$
>
> What are the differences with reward shifting in ReNorm? In your derivation, (r - c) directly equate to the TD-error (\delta - c). Please rigorously derived the change in V. If 'the value function can certainly change and adapt to this shift and cannot fully absorb this shift', how does your derivation differ from the presented above?
>
> It is highly probable that the superior results​ in your experiments are due to the function of $ \hat{r} $, not ReNorm. Therefore, I argue that this paper claim regarding ReNorm's contribution is incorrect.

---

> > ### Author Response · Authors · 2025-11-22
> >
> > Thanks for the detailed derivation — We agree that, under the assumptions you are using, a constant reward shift leaves GAE invariant. Your derivation corresponds to the infinite-horizon MDP case with a critic that has already re-converged to the new Bellman fixed point, where the value function can shift by a single global constant and exactly absorb the reward offset.
> >
> > The key difference is that our setting is not the infinite-horizon fixed-point case:
> >
> > 1. **We use finite-horizon, truncated GAE.**
> > In this regime, the correct transformed value function is time-dependent (its offset depends on the remaining horizon). A single state-only function $V(s)$ cannot represent this family of offsets. Even if the critic re-trains, there is no single constant the critic can add that simultaneously matches all timesteps.
> >
> > 2. **Process rewards are applied turn-wise and trajectories have variable length.**
> > Subtracting a “constant per reward” does not translate into a constant per advantage; instead it induces a time-dependent shift whose magnitude depends on how many steps remain in the episode. Two trajectories with different numbers of turns experience different effective offsets. This creates advantage distortions even if the state distributions are similar.
> >
> > So the difference between your derivation and ours is not in the algebra, but in the assumptions.
> >
> > ### Derivations in change of V:
> >
> > We work with finite-length trajectories of terminal time $T$, with truncated GAE:
> >
> > $$
> > A\_t = \sum_{l=0}^{T-t-1} (\gamma \lambda)^l \delta\_{t+l}, \quad V(s\_T) = 0.
> > $$
> >
> > In the finite-horizon case, the correct value function is time-indexed:
> >
> > $$
> > V\_t(s) := \mathbb{E}\left[ \sum_{k=0}^{T-t-1} \gamma^k r\_{t+k} \\Big|\ s\_t = s \right].
> > $$
> >
> > Now we apply the same constant reward shift $r^{\prime}\_t = r\_t - c$ and define:
> >
> > $$
> > V^{\prime}\_t(s) := \mathbb{E}\left[ \sum_{k=0}^{T-t-1} \gamma^k r^{\prime}\_{t+k} \\Big|\ s\_t = s \right].
> > $$
> >
> > Compute the change in $V$ explicitly. First, substitute the shifted reward:
> >
> > $$
> > V^{\prime}\_t(s) = \mathbb{E}\left[ \sum_{k=0}^{T-t-1} \gamma^k (r\_{t+k} - c) \\Big|\ s\_t = s \right]
> > $$
> >
> > Separate the terms:
> >
> > $$
> > V^{\prime}\_t(s) = \mathbb{E}\left[ \sum_{k=0}^{T-t-1} \gamma^k r\_{t+k} \\Big|\ s\_t = s \right] - c \sum_{k=0}^{T-t-1} \gamma^k
> > $$
> >
> > Substitute $V\_t(s)$ back in:
> >
> > $$
> > V^{\prime}\_t(s) = V\_t(s) - c\ G(t)
> > $$
> >
> > where the geometric sum $G(t)$ is:
> >
> > $$
> > G(t) := \sum_{k=0}^{T-t-1} \gamma^k = \frac{1 - \gamma^{T-t}}{1 - \gamma}.
> > $$
> >
> > So in the finite-horizon case, the change in the value function is:
> >
> > $$
> > V^{\prime}\_t(s) = V\_t(s) - c\ G(t),
> > $$
> >
> > with $G(t)$ depending on the remaining horizon $T - t$.

---

> ### Comment · Reviewer_FRDx · 2025-11-22
>
> According to your derivation and calculate the new td-error:
>
> $$ \delta^{\prime}_t = {r_t^{\prime}} + \gamma V^{\prime} (s\_{t+1}) - V^{\prime} (s_t) $$
>
> Let $ r^{\prime}_{t} = r_t - c$
>
> $$ {\delta}^{\prime}_t = (r\_{t} - c) + \gamma \left[ V (s\_{t+1}) - cG(t+1)\right] -  \left[ V (s_t) - cG(t) \right] $$
>
> Then we have
>
> $$ {\delta}^{\prime}_t  =  {\delta}_t + c \left[ -1 -\gamma G(t+1) + G(t) \right] $$
>
> Notice that
>
> $$ -1 -\gamma G(t+1) + G(t) = -1 + \frac{1 - \gamma^{T-t} - \gamma(1 - \gamma T^{T-t-1})}{1-\gamma} = -1 + \frac{1-\gamma}{1-\gamma} = 0$$
>
> Then td-error is also invariant:
> $$ {\delta}^{\prime}_t  = {\delta}_t $$
>
> Unless your G(t) in the calculation of GAE is time-independent term, or t will reset the shift to zero.

---

> > ### Author Response · Authors · 2025-11-25
> >
> > Substituting this form into GAE implicitly **assumes that the critic has already re-converged to the new Bellman fixed point**. However, during training $G(t)$ changes and varies across t steps, so this assumption does not always hold and the theoretical equivalence breaks down. Therefore, this shifted value formulation **cannot** be directly applied to the GAE formulation during the dynamic training process.
> >
> > Our derivation of \$V'(S_t)=V(S_t)-cG(t)\$ was intended to show that introducing the shift makes the value function explicitly time-dependent. Since the value function is predicted by an LLM, **it is challenging for the model to continuously adjust and accurately track the target whenever $G(t)$ changes, and correctly represents this family of offsets**. Thus, introducing this shift does affect the training process.

---

> ### Comment · Reviewer_FRDx · 2025-11-25
>
> I agree with your point that the reward shift $c$ alters the Bellman fixed point of the value function, and requires the value network to reconverge. However, within the PPO-GAE, once the value network adapts to this new fixed point through training, the resulting offset cancels out when computing the advantage function Adv. The core of GAE is based on td-error, which itself is invariant to a constant reward shift. Therefore, this does not affect the unbiased property of the policy gradient estimation. In practical, we usually stabilize training through post-processing like advantage normalization, rather than directly shifting the raw rewards by a constant $c$.
>
> Furthermore, if the dynamic optimization process fundamentally deviates from the theoretical assumptions, what meaningful basis does any analysis based on PPO have? Such point, if valid, would undermine the theoretical basis upon which the algorithm operates.
>
> Finally, I believe that omitting the ReNorm trick in this work would not compromise the experimental results, while also avoiding theoretical flaws. Highlighting the effectiveness of ReNorm in the contribution section would be superfluous.

---

> ### Author Response · Authors · 2025-11-26
>
> We are glad that we all agree the introduced shift affects the training dynamics, and therefore has an impact on the experimental outcomes.
>
> The common agreement is that, due to the time-dependent term, the value function cannot immediately adapt to the new convergent point. As a result, when the time-dependent term changes (i.e., when the action-turn changes under our step-level credit assignment), the reward shifts while the value function lags behind. This discrepancy temporarily alters the advantage estimates and leads to corresponding adjustments during optimization. Our analysis is conducted under the standard and reasonable assumptions of PPO, and focuses on how the algorithm behaves in practical settings. Examining the gap between ideal assumptions and real-world implementation does not make the analysis invalid (e.g. critic function itself is a function approximation, but remain within the PPO framework and is considered a reasonable deviation from the ideal case).
>
> To assess the effectiveness of ReNorm, we have compared it against several normalization techniques (e.g., rescaled PRM, dynamic tuning) and perform ablation studies (e.g., without normalization, without shift). All results consistently show that ReNorm stabilizes training and improves performance. We also observe different behavior across action turns when using different normalization methods (i.e., with/without shift, Appendix A.9.2). Therefore, we respectfully disagree with the claim that ReNorm does not influence the experimental results—our empirical evidence clearly demonstrates otherwise.
>
> We truly appreciate the discussions and your comments, and will include this expanded theoretical explanation in the revised version to help readers better understand our design. However, this theoretical discussion we provide is meant to offer insight into why the shift affects training dynamics as **one part of** the ReNorm mechanisms, served as **additional analysis**. It does not change the experimental results or contributions, and our claims and performance remain clear and fully supported.
>
> We hope this clarification addresses your concerns.

---

### Official Review · Reviewer_9HGM · 2025-11-04

**Soundness:** 3
**Presentation:** 3
**Contribution:** 1
**Rating:** 4
**Confidence:** 3

**Summary:**

This paper proposes a type of process reward for training LLM search agents with RL. Specifically, the authors propose of process reward they call "Principle Process Reward", which is a (prompted) reward derived from a set of prespecified set of "principles" (e.g., covering aspects like formatting, correctness, etc). Like other process reward work, this is combined with the outcome reward, but with a particular type of heuristic normalization applied (called Reward Normalization). The paper demonstrates that this method outperforms non-RL baselines and other RL methods (both outcome- and process-based) through experiments on several QA benchmarks.

**Strengths:**

The overall approach is sensible and well-evaluated. Using a generated reward model to provide rubric-based feedback is a practical way to get dense reward signal for a search agent. The ReNorm strategy is also simple and intuitive. Experimental setup appears sound, and it seems clear that this process reward leads to better performance.

**Weaknesses:**

The main weakness of this work is novelty and technical contribution. Using process rewards is well established in the literature. Generative rubric-based reward models have also been explored before, see e.g., Gunjal et al 2025, Anugraha et al 2025, or generative reasoning-based rewards more broadly, e.g., Guo et al 2025. While the resulting system works well, the paper primarily demonstrates that a well-engineered combination of known components (generative PRM + outcome reward + a simple normalization rule) is effective for training search agents. This is a useful finding, but the lack of substantial novel insight (and unclear implications as to its broader applicability, see questions below) makes it a borderline case for ICLR in my opinion.


[1] Gunjal et al 2025. https://arxiv.org/pdf/2507.17746v1

[2] Anugraha et al 2025. https://arxiv.org/pdf/2505.13388v1

[3] Guo et al 2025. https://arxiv.org/pdf/2505.14674

**Questions:**

- I'm having a hard time understanding what the <max_score> exactly is? An upper bound on the score? I don't see it defined anywhere.
- How was the set of principles developed and validated? It is unclear how generalizable it is, or if it would need to be re-engineered for different types tasks.
- Renorm is presented as a core contribution --- I'm curious as to how this compares to standard methods for combing rewards like some convex combination of $r_p$ and $r_o$ or using the process reward as a potential function.

---

> ### Author Response · Authors · 2025-11-20
>
> We appreciate your valuable feedback. We will incorporate the clarifications and address the issues in the next draft. Here is our response to your questions:
>
> ---
>
> > The overall approach is sensible and well-evaluated. But the main weakness of this work is novelty and technical contribution.
>
> We thank the reviewer for acknowledging the clarity of our motivation and the strength of our empirical evaluation. We agree that generative rubric / reasoning–based reward modeling is an active area and promising direction, as reflected in the concurrent works [1–3]. **However**, our contributions and novelty differ substantially from these studies.
> Unlike [2] and [3], which primarily evaluate the **accuracy of GenRM-produced rewards**, our work investigates whether a principled process reward model can serve as **effective and stable supervision during full RL training**. Unlike [1], which uses GenRM mainly as an outcome reward on in-house datasets, we focus on the more challenging setting of **non-verifiable process rewards**, and study how to design and apply principled criteria for multi-turn tool-use agentic RL, validated and compared on public search-agent benchmarks. Moreover, we introduce **ReNorm**, a new method for unifying process and outcome rewards, based on token level credit assignment for process rewards. While its design is intuitive, it is far from trivial—we provide extensive comparison with alternative normalization schemes (Table 3, Appendix. A.9.1) and gradient-based evidence showing its stabilizing training effect (Appendix. A.9.4).
> Overall, our paper offers new insights into principled process reward design and its practical benefits for non-verifiable multi-step RL training, which is the **first** in agentic RL domain, and prove the effectiveness of our method through extensive experiments.
>
>
> > I'm having a hard time understanding what the [object Object] exactly is? An upper bound on the score? I don't see it defined anywhere.
>
> Thank you for the question. We are not sure what is “[object Object]”. Could you please clarify it?
>
> > How was the set of principles developed and validated? It is unclear how generalizable it is, or if it would need to be re-engineered for different types tasks.
>
> The principles used in our method are human-designed general guidelines that serve as reference criteria for the LLM to evaluate process steps. These principles were iteratively developed and refined through repeated human assessment and validation. We have verified the effectiveness of the principles across multiple QA benchmarks within the search task domain, showing the generalizability different types of search-based reasoning tasks. For tasks outside the search domain, task-specific principles may require adjustment. However, many core principles, such as relevance to the original query and whether it is a redundant action, are broadly applicable and substantially reduce the effort for re-engineering, making our approach easy to adapt to new domains.
>
> > Renorm is presented as a core contribution --- I'm curious as to how this compares to standard methods for combing rewards like some convex combination of rp and ro or using the process reward as a potential function.
>
> We thank the reviewer for the suggestion. We presented the ReNorm is one of the core contribution of our work, and have done various comparison with different normalziation methods to verify its effectiveness, including (1) directly assign raw process rewards at each step (2) scaling the sum of process rewards in a trajectory to [0,1] (3) directly summation of all process rewards in a trajectory (See Table 3). We additionally conducted experiments for convex combination of $r_p$ and $r_o$ in the formulation of $R=2*(m*r_p+(1-m)*r_o)-1$, in a setting of training qwen2.5-3b in NQ dataset (more details at Appendix A.9.1)
>
> | methods/benchmark           | NQ    | TriviaQA | PopQA |
> |-------------|-------|----------|-------|
> | 0.25        | 0.219 | 0.315    | 0.216 |
> | 0.75        | 0.343 | 0.507    | 0.390 |
> | **Ours** | **0.423** | **0.565**    | **0.411** |
>
>
> As the result shows, our setting achieves a balance in $r_p$ and $r_o$, with both increasing in training stability and performance. Inbalance settings may introduce noise and cause training collapse.
> For potential function, the closest work we found in LLM RL training with reward potential function is Chan et al. [4], which constructs token-level dense rewards using attention from a transformer-based reward model and shows this redistribution is equivalent to a potential function. However, their approach **requires a large transformer RM with accessible attention weights**, which are not available in our current setting. Furthermore, we would like to emphasize that our proposed method can introduce richer step-level feedback, beyond the capability of potential-based reward formulations with relying on 0/1 outcome reward.
>
> [4] Chan, et al, 2024 https://arxiv.org/pdf/2402.00782

---

> ### Comment · Reviewer_9HGM · 2025-11-20
> **Clarification**
>
> Thanks for the reply --- [object Object] is weird, sorry about that: I meant to copy paste the "\<max_score\>" term from the paper.

---

> > ### Author Response · Authors · 2025-11-21
> >
> > Thanks for your clarification. Yes, it is the upper bound on the score, where we need to unify the process and outcome reward into the same scale. We will update with more clear definition in the paper.

---

### Author Response · Authors · 2025-11-20
**Summary of Updates Based on Reviewers' Feedback**

We thank all reviewers for the comments and suggestions! We have revised the paper to address the main concerns, with updates highlighted in **blue** in **Appendix A.9**.

Specifically, we add:
- An analysis of different aggregation strategies between $r_p$ and $r_o$ (A.9.1).
- An analysis of the effect of constant shift in ReNorm (A.9.2).
- A discussion on dynamically tuning the positive/negative ratio in ReNorm (A.9.3).
- More metrics to indicate our method effectiveness in stable RL training (A.9.4).
- An extend theoretical analysis of ReNorm (A.9.5).

We hope these revisions adequately address the reviewers’ concerns and looking forward to replies and further discussion.

---

### Meta-Review · Area_Chair_LPaj · 2025-12-16

**Summary:**

Reviewer 9HGM recommended borderline reject. Their main concern was the lack of technical novelty, since process reward models have been explored in multiple prior works, and the reward normalization scheme, which is quite simple, seems like a simple contribution by comparison. They also brought up the concern that the process reward models designed for search may not generalize to other settings.

Reviewer FRDx recommended strong reject. They raised the concern that the core contribution (reward normalization) was overly simplistic, and that the stated reasons did not account for the improvements and performance. They had a lengthy discussion with the authors over the correctness of one of the core technical claims, which was not resolved after multiple back-and-forth.

Reviewer ENAr was the most positive and recommended borderline except. They noted the paper's solid and empirical evaluation, while also noting the limited novelty and unclear applicability beyond the considered experimental settings as the other reviewers.

Overall, I agree with all reviewers that the technical novelty and applicability beyond the considered settings seems limited for an ICLR paper. I also agree with reviewer FRDx that the reward normalization, which is claimed as one of the core contributions, may not be helping for the reasons stated in the paper (see my comment below). Based on all of these, and the fact that no reviewer seemed likely to champion the paper, I am recommending reject.

**Reviewer Concerns:**

I think reviewer 9HGM's concerns, namely, about the technical novelty and applicability beyond the considered settings, probably remain even after the authors rebuttal. Overall, I agree with these concerns: the paper seems technically solid, but incremental for ICLR.

It seems unlikely reviewer FRDx would have changed their score past 4, based on the fact that their concerns persisted even after a lengthy back-and-forth. This does not necessarily mean that their criticisms are true, but I agree that the simple reward normalization (simply subtracting 1) is unsatisfying. This seems like an ad-hoc way of preventing the model from trying to achieve high reward by simply extending the length of the trajectories, which can happen since the process rewards are all non-negative. Incidentally, this is a well documented phenomenon in (non-LLM) RL and their exist principled solutions to the problem--for example, ensuring all shaped rewards (process rewards here) sum to 0 [1].

The authors addressed some of reviewer ENAr's concerns in their rebuttal. However, these were mostly clarifying questions and cosmetic changes, and I don't think the main concern about technical novelty and applicability beyond search would have changed.



[1] Policy invariance under reward transformations: Theory and application to reward shaping. Ng et al, ICML 1999.

**Reviewer Scores:**

R1: no change
R2: either no change, or 2-> 4.
R3: probably no change.

---

### Decision · Program_Chairs · 2026-01-26

Reject